# Shape-recovery of implanted shape-memory devices remotely triggered via image-guided ultrasound heating

Yang Zhu [1,2,3,4,5,16] ✉, Kaicheng Deng[1,16], Jianwei Zhou[6,16], Chong Lai[7,16], Zuwei Ma[2], Hua Zhang[8], Jiazhen Pan[8], Liyin Shen[1], Matthew D. Bucknor[9], Eugene Ozhinsky [9], Seungil Kim [2,10], Guangjie Chen[11], Sang-ho Ye[2,10], Yue Zhang[12], Donghong Liu[13], Changyou Gao [1], Yonghua Xu[14] ✉, Huanan Wang[8] ✉ & William R. Wagner [2,3,10,15] ✉

Shape-memory materials hold great potential to impart medical devices with functionalities useful during implantation, locomotion, drug delivery, and removal. However, their clinical translation is limited by a lack of non-invasive and precise methods to trigger and control the shape recovery, especially for devices implanted in deep tissues. In this study, the application of image-guided high-intensity focused ultrasound (HIFU) heating is tested. Magnetic resonance-guided HIFU triggered shape-recovery of a device made of poly-urethane urea while monitoring its temperature by magnetic resonance ther-mometry. Deformation of the polyurethane urea in a live canine bladder (5 cm deep) is achieved with 8 seconds of ultrasound-guided HIFU with millimeter resolution energy focus. Tissue sections show no hyperthermic tissue injury. A conceptual application in ureteral stent shape-recovery reduces removal resistance. In conclusion, image-guided HIFU demonstrates deep energy penetration, safety and speed.

A wide variety of shape memory biomaterials can maintain their temporary shape at physiological conditions and undergo rapid geometrical changes when triggered, usually through heat-induced phase transition[1,2]. Applying these materials and this functionality to medical devices would be valuable in a variety of clinical scenarios, thus the concept has long attracted researchers in related fields[3–5]. Specifically, applications of shape memory biomaterials and devices in minimally invasive delivery of functional tissues[6], embolization of aneurysms[7],

[1]MOE Key Laboratory of Macromolecular Synthesis and Functionalization, Department of Polymer Science and Engineering, Zhejiang University, Hangzhou, Zhejiang, China. [2]McGowan Institute for Regenerative Medicine, University of Pittsburgh, Pittsburgh, PA, USA. [3]Department of Bioengineering, University of Pittsburgh, Pittsburgh, PA, USA. [4]Binjiang Institute of Zhejiang University, Hangzhou, China. [5]Key Laboratory of Cardiovascular Intervention and Regenerative Medicine of Zhejiang Province, Sir Run Run Shaw Hospital, Zhejiang University School of Medicine, Hangzhou, China. [6]School of Electromechanical and Energy Engineering, NingboTech University, Ningbo, Zhejiang, China. [7]Department of Urology, First Affiliated Hospital, School of Medicine, Zhejiang University, Hangzhou, Zhejiang, China. [8]College of Animal Sciences, Zhejiang University, Hangzhou, Zhejiang, China. [9]Radiology and Biomedical Imaging, University of California, San Francisco, CA, USA. [10]Department of Surgery, University of Pittsburgh, Pittsburgh, PA, USA. [11]Department of Urology, The Children's Hospital, School of Medicine, National Clinical Research Center for Child Health, Zhejiang University, Hangzhou, Zhejiang, China. [12]San Francisco Veterans Affairs Medical Center, University of California, San Francisco, CA, USA. [13]College of Biosystems Engineering and Food Science, National-Local Joint Engineering Laboratory of Intelligent Food Technology and Equipment, Zhejiang Key Laboratory for Agro-Food Processing, Zhejiang University, Hangzhou, Zhejiang, China. [14]Department of Imaging and Interventional Radiology, Zhongshan-Xuhui Hospital of Fudan University/Shanghai Xuhui Central Hospital, Shanghai, China. [15]Department of Chemical Engineering, University of Pittsburgh, Pittsburgh, PA, USA. [16]These authors contributed equally: Yang Zhu, Kaicheng Deng, Jianwei Zhou, Chong Lai. ✉e-mail: zhuyang@zju.edu.cn; howardyonghua@yeah.net; hnwang@zju.edu.cn; wagnerwr@upmc.edu

drug release[8], device implantation[9], and self-tightening sutures[10] have been reported.

In circumstances where the device is designed to change shape after in vivo placement, it is desirable to trigger shape recovery non-invasively to minimize associated medical risks[11,12]. Remote heating meets this requirement as: (1) heat-induced phase transition is the most commonly adopted mechanism to trigger shape change of shape memory polymers and metals, and (2) heating energy could be non-invasively transferred to biomaterials implanted underneath multiple tissue layers by electromagnetic or mechanical waves[13]. For remote heating of implanted shape memory devices to be clinically accep-table, it is important that the heating energy is focused on the bio-material and is briefly applied to avoid hyperthermic injury of surrounding tissues. Heat triggers such as certain wavelength windows of light and alternating magnetic fields have been extensively studied and proven effective in triggering biomaterial shape memory[14–16]. However, optical energy dissipates and attenuates quickly in human tissue, which significantly lowers the efficiency of targeted heating and increases the hyperthermia risk in tissues on the light path[17]. Alter-nating magnetic fields heat by induction, but the same heating mechanism is associated with risks of "microwaving" tissues covered in the magnetic field[18]. These risks have become obstacles to the trans-lation of shape memory biomaterials. To date, only a few studies have reached the small animal stage[19]. Remote heating of shape memory biomaterials in large animal models has not yet been reported.

Compared to the two heating options above, high-intensity focused ultrasound (HIFU) concentrates ultrasound energy on in vivo targets for efficient heating, while less energy is attenuated in surrounding tissue. In fact, focused ultrasound including HIFU has been widely used, and its safety and efficacy in various applications

have been demonstrated[20–23]. In addition, with nearly real-time mag-netic resonance or ultrasound guidance (MR guided HIFU, MRgHIFU; US guided HIFU, USgHIFU), the sound wave energy can be more precisely targeted to minimize off-target damage[24,25]. Supported by automated 3D scanning at millimeter level precision, HIFU heating on sophisticated paths and large volumes can be programmed. In MRgHIFU, local temperature can be monitored by MR thermometry[26]. Given these advantages, HIFU has been FDA approved for tumor ablation use[20]. Triggering material shape recovery with HIFU has been demonstrated as versatile and precise in vitro[27,28]. Therefore, image-guided HIFU is an attractive trigger mechanism that may overcome existing barriers to the comprehensive clinical translation strategy of shape memory devices (Fig. 1A, B).

We hypothesized that image-guided HIFU could be a safe, effi-cient, non-invasive heating solution for triggering shape recovery of shape memory medical devices. To test this hypothesis, we fabricated a series of shape memory polyurethane ureas (PUUs) with tunable transition temperatures and demonstrated the feasibility of remotely triggering precise shape change of PUU with MRgHIFU. A demon-stration of rapid shape recovery of a device in a canine bladder trig-gered by USgHIFU was performed and visualized in vivo. Examination of surrounding tissues and finite element simulations were used to evaluate the safety of the procedure.

## Results
### PUU transition temperature control
The structure of a PUU with poly(caprolactone) soft segment (PUU-PCL) (Fig. 2A) was confirmed by [1]H-NMR as shown in Figure S1. [1]H peaks characteristic of the soft segments and characteristic peaks from urea and urethane groups of the hard segment can be found in Fig. S1.

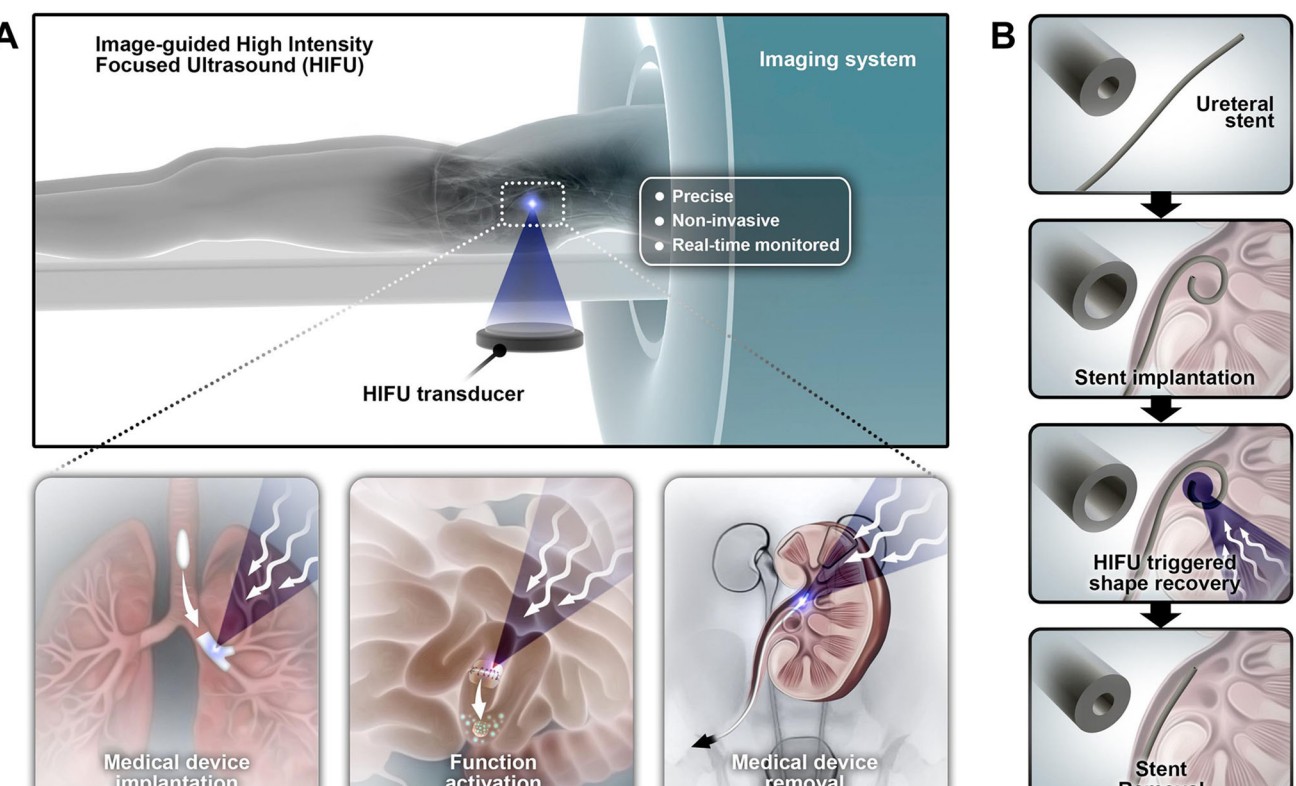

**Fig. 1 | Scheme of remotely triggered in vivo shape recovery of shape memory medical devices by image-guided HIFU and the potential clinical applications. A** The equipment layout of image-guided HIFU. Image-guided HIFU supports pre-cise, non-invasive, real-time monitored heating of shape memory medical devices in vivo, which could be used in medical device implantation, function activation including drug release, and device removal. **B** Concept of shape memory ureteral stent removal. Image-guided HIFU triggers the diameter decrease and straighten-ing of the J shaped coil of the stent in the renal pelvis, which lowers the resistance from the urethral and adjacent soft tissues in stent removal.

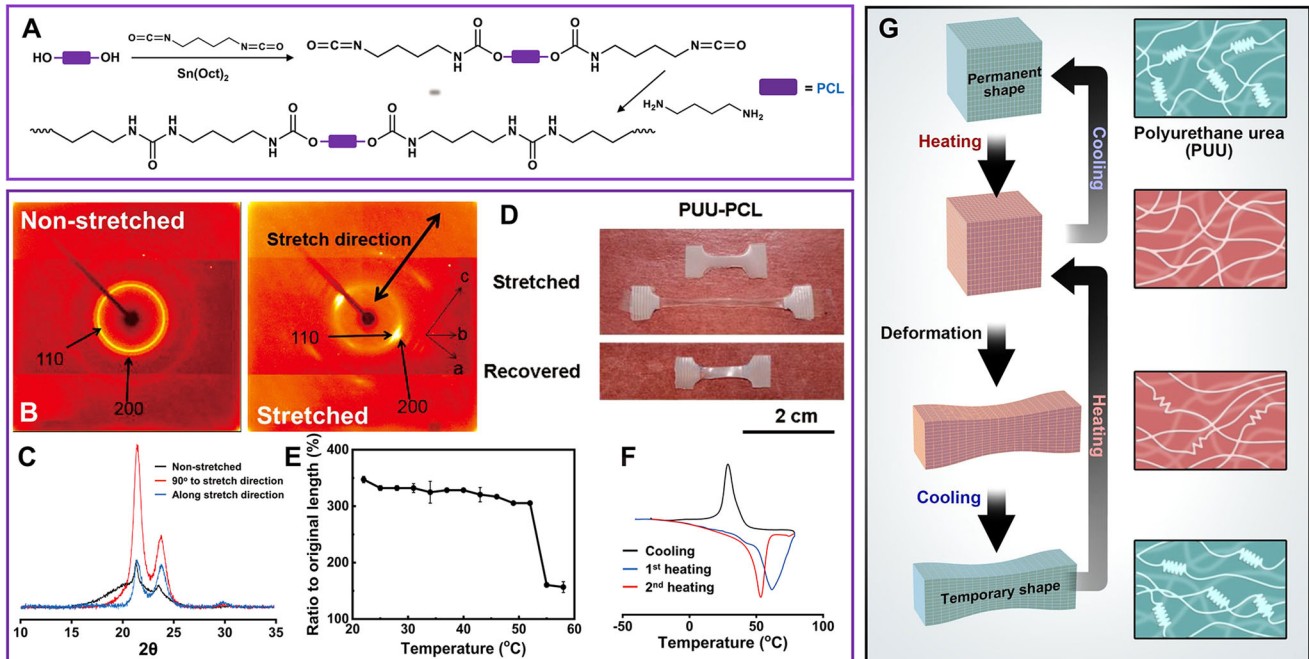

**Fig. 2 | Shape memory property of PUU-PCL. A** Structure and synthesis route of PUU-PCL. In which PCL is the soft segment, and its crystallization maintains the temporary shape. **B** 2-D XRD spectra of non-stretched and stretched PUU-PCL. The peaks 90° to the stretch direction became stronger after stretching, which indicated the enhanced crystallization. **C** 1-D XRD spectra of non-stretched and stretched PUU-PCL. **D** Appearances of pristine, stretched, and recovered PUU-PCL samples. Dumbbell PUU-PCL samples were stretched to 350% original length and recovered. (**E**) Sample length measured at different temperature, which gives the transition temperature ($n = 3$ per group), data is presented as means ± SD. **F** DSC curve of PUU-PCL. The melting temperature is 53 °C at 2nd heating cycle, which is consistent with the result in (**E**). **G** Mechanism of the shape memory.

Stretch-induced molecular orientation and re-crystallization of PUU-PCL was investigated by wide angle XRD. In the 2-D XRD patterns of the non-stretched PUU-PCL, (Fig. 2B), a broad circle corresponding to 110 and 200 lattice planes of PCL can be observed. In the stretched PUU-PCL, the circle pattern transferred into discontinuous, symmetrical arcs, indicating stretch-induced molecular orientation and re-crystallization[29]. On the 1-D XRD spectrum (CuKα) for the non-stretched sample, peaks at $2\theta = 21.6°$ and $24°$ were observed, which were consistent with the diffraction of the 110 and 200 lattice planes of orthorhombic crystalline PCL (Fig. 2C). In the stretched samples, the peaks corresponding to 90° to the stretched direction become significantly stronger as a result of stretch enhanced crystallization. In contrast, the peaks of the stretched direction were weaker compared to the ones of the non-stretched samples.

The PUU-PCL samples can be stretched to >500% strain, and exhibited classical cold drawing phenomenon during stretching (Fig. 2D, Fig. S2). The temporary shapes of PUU-PCL devices were fixed instantly (<1 s) after being deformed at room temperature, or cooled in cold water after deformed above PCL melting temperature. The stretch induced crystalline domains can be melted in 53 °C saline, resulting in instant shape recovery of the samples, leaving about 50% permanent shape change (Fig. 2C–E). The measured transition temperature (53°C) agreed with the melting peak of the 2nd heating on DSC (Fig. 2F), which confirmed the crystallization-based shape memory mechanism (Fig. 2G). Copolymerization with valerolactone (VL) or poly(ethylene glycol) (PEG), or adding $Fe_3O_4$ nanoparticles both lowered the melting temperature (Fig. S3). The transition temperatures of PUU-PEGPCL and PUU-PVLCL were lower than 37 °C and thus deemed not suitable for in vivo use. Addition of 30% (w/w) $Fe_3O_4$ nanoparticles into PUU-PCL lowered the melting temperature to 47 °C (Fig. 2E), and 47 °C was set as the target temperature in HIFU experiments.

## Image-guided HIFU triggered shape recovery in vitro

PUU-PCL was fabricated into an 8-finger palm and 4 of the fingers were folded to 90° (Fig. 3A). Each finger was 0.5 cm wide. The PUU-PCL palm appeared dark under MRI due to its low water content. The resolution of MRI was high enough to differentiate the 3 different types of cross-sections of the palm – the "∟" shape of the folded fingers, the "_" shape of the non-folded fingers, and the "⊥" shape on the boundary of the folded and non-folded fingers, as shown in Fig. 3B–E. MRgHIFU successfully targeted the PUU-PCL samples positioned along the exposed surface of the gel pad and focused the ultrasound energy on the desired locations. Each individual sonication was prescribed at approximately 3000 J sonication with heating time of 20 s (150 W acoustic power) and frequency of 1.05 MHz. The ultrasound energy was focused on the border of 2nd and 3rd fingers. MRgHIFU heating triggered unfolding of the 2nd finger in 30 s while other fingers including the 1 cm distant 4th finger were unaffected (Fig. 3F). A minor color change could be observed around the focus (Fig. 3F), possibly due to crystallinity change after the heating/cooling cycle.

MR thermometry demonstrated a change in the gel pad temperature adjacent to the sample with a standard proton resonance frequency shift method. A snapshot of the continuous temperature monitoring showed that the heating primarily occurred in a 1 cm diameter sphere around the focus (Fig. 3G). The local temperature was followed throughout the entire heating/cooling cycle, as shown in Fig. 3H. By tuning the HIFU power, the target region of PUU-PCL was heated just above its transition temperature (53 °C), maintained for ~10 s and cooled (Fig. 3H). Higher powers resulted in significantly higher temperatures within the same heating time. In addition, MRgHIFU with temperature monitor allowed one to stop heating at any time to prevent overheating.

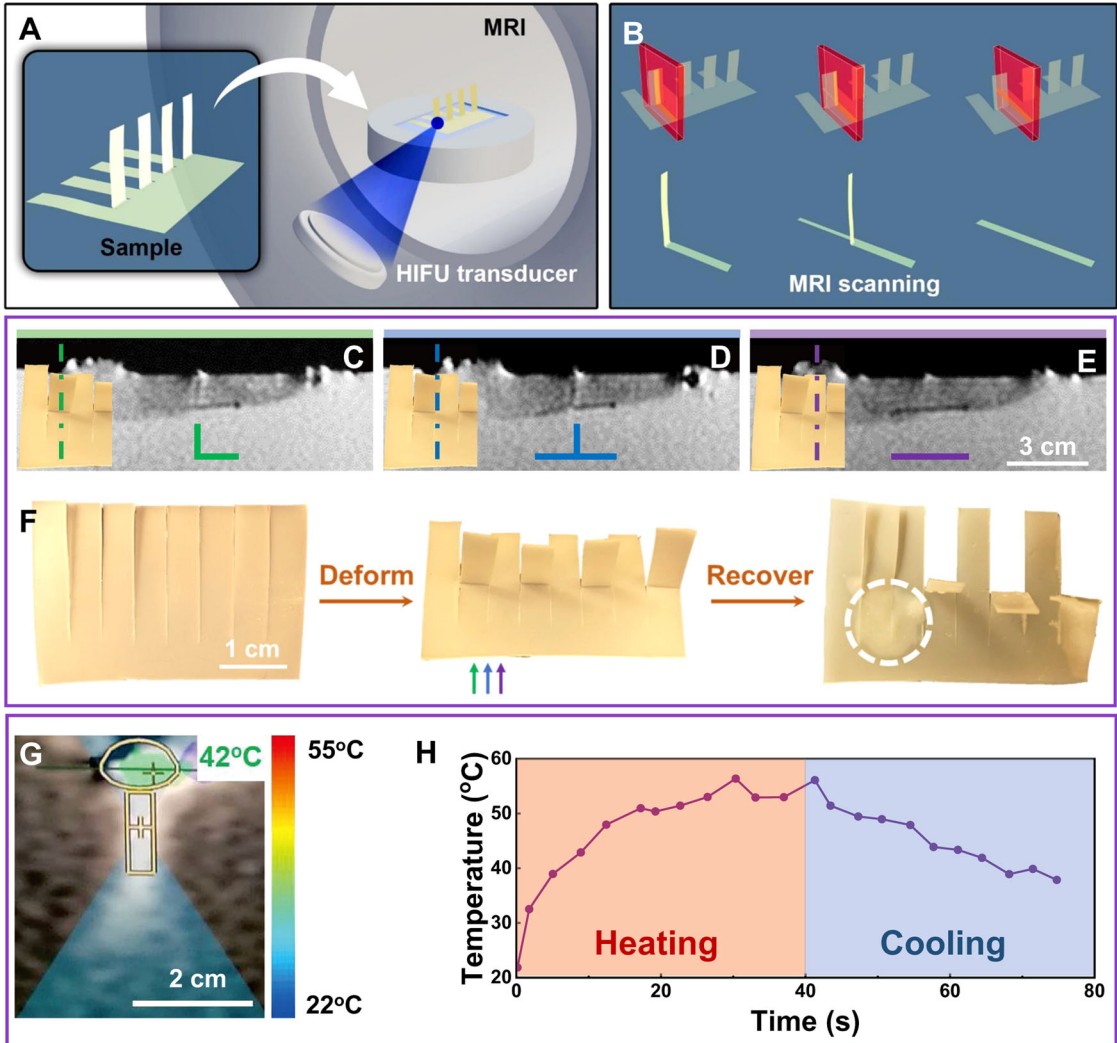

**Fig. 3 | In vitro heating of PUU-PCL triggered by magnetic resonance-guided HIFU (MRgHIFU). A** Experiment set up of MRgHIFU heating of the palm-shaped PUU-PCL device. Agarose gel pads as tissue phantoms were placed on the table directly above the transducer and a 1 cm depression was cut and filled with ultra-sound gel and water, immersing the PUU-PCL samples on which HIFU energy was focused. **B** cross-sections of the palm-shaped device, **C** the "⌐" shape of the folded fingers, **D** the "⊥" shape on the boundary of the folded and non-folded fingers, and **E** the "_" shape of the non-folded fingers. **F** Transformation of the palm-shaped device between its permanent shape and temporary shape. The white circle shows the HIFU heated point. Colored arrows show the positions of the cross-sections of in (**B-D**). **G** Local temperature measurement during heating shown on the user interface of MRgHIFU. The yellow circle shows the interface of PUU-PCL and the gel pad, where the temperature was measured. The yellow rectangle and cross show the focal point of HIFU. **H** Measured temperature of the focal point on the palm-shaped device during heating and cooling.

## Conceptual shape recovery and removal of ureteral stents ex vivo

Double-J ureteral stents are used to restore urine flow from the kidney when ureters are obstructed by pathological complications including kidney stones and tumor-related compression. Stents may also be placed in a ureter to protect the urine pathway irritated or damaged during an ureteroscopy procedure. However, the stent removal procedure is highly uncomfortable for patients and carries bleeding and infection risks. Straightening the J shaped coil that resides in the kidney and decreasing the stent diameter prior to removal could minimize damage to epithelial tissue of the ureter and lower the risks of associated complications.

PUU-PCL was cast into a straight tube with an outer diameter smaller than commercial products (permanent shape). $Fe_3O_4$ nanoparticles were added to the polymer substrate to lower the transition temperature and increase MR contrast. The nanoparticle-incorporated tubes were coiled into single-J stents and expanded to match their diameter to commercial products, as shown in Fig. 4A. Stents in the

temporary shape could swiftly recover to the permanent shape (uncoil of the single-J part and decrease in diameter within 3 s, as shown in Supplementary Movie 1). Fast recovery was also achieved in other tests. Sealed tubes reopened and released the loaded dye upon heating in 55 °C water in less than 3 s, as well (Supplementary Movie 2). Laser targeted heating recovered the permanent shape of a PUU-PCL tube (visualized mimicking of HIFU heating) in <10 s (for each folded end, as shown in Supplementary Movie 3). In an ex vivo stent removal assessment, stents were pulled through a porcine ureter (Fig. 4B). For a commercially available stent, the load increases as the J shaped coil and main body straightens, and gradually decreases as the stent pass through the ureter (Fig. 4C). The elastic stent tended to restoration of its original shape, therefore the uncoiled J structure pressed the inner surface of the ureter and significantly deformed the latter. The peak load and force required to slide the stent through the ureter with the J-coil removed from commercially available stents was significantly decreased compared to the original ones, showing that the J-coil was a primary contributor to the resistance encountered with stent removal

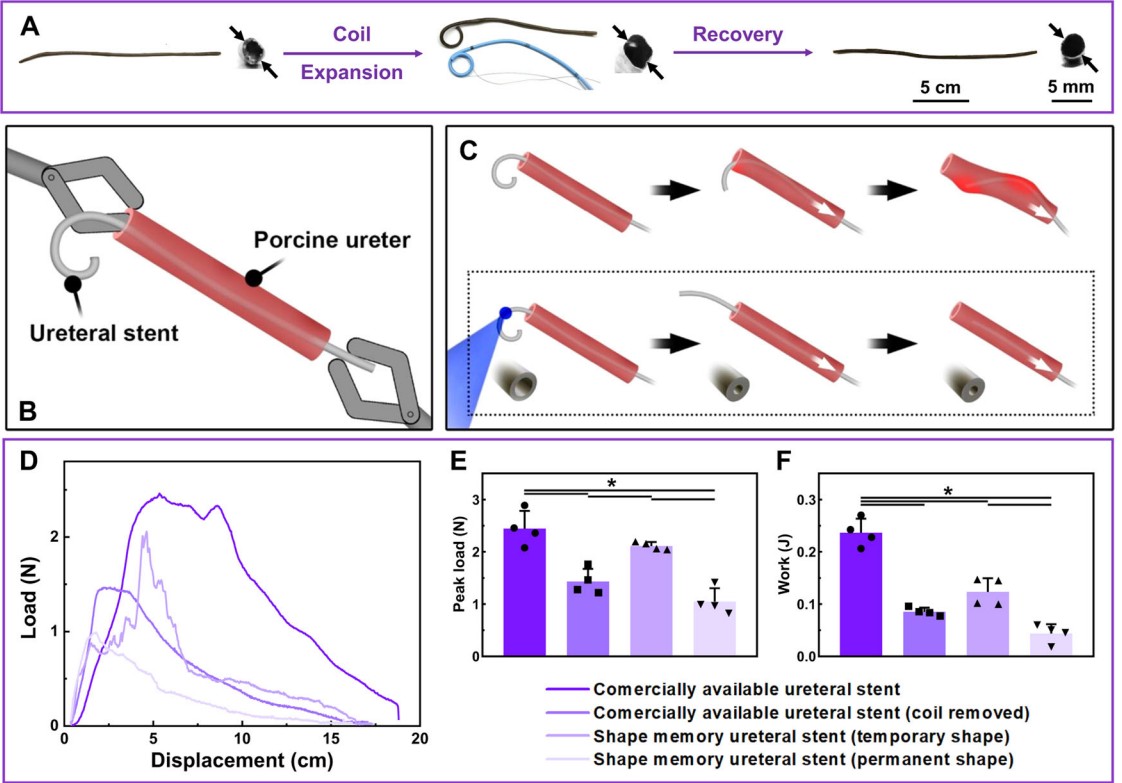

**Fig. 4 | Ex vivo ureteral stent removal demonstration. A** One end of the $Fe_3O_4$ nanoparticle-incorporated PUU-PCL stent was coiled, and the diameter of stent was increased by balloon expansion. The J shaped coil was straightened, and the stent shrank to its original diameter when triggered at high temperature. **B** Scheme of ex vivo stent removal from a porcine ureter, including mimicry of J shaped coil withdrawal from renal pelvis. **C** Resistance analysis of ureteral stent removal from the porcine ureter. Upper row: the elastic J shaped coil mimicking a commercial device compresses the inner surface of the ureter, generating friction during device removal. Lower row: HIFU triggers shape recovery of PUU-PCL stent, decreasing

removal resistance. **D** Resistance measured during removal of ureteral stents from the porcine ureter. Resistance increases as the pendant end of the stent enters the ureter and decreases as the contact between the ureter and stents decreases. Peak load (**E**) and work (**F**) required to remove the stents were higher in commercially available stents and PUU-PCL stents with coils compared to their coil-removed counterparts ($n = 4$ per group). Coil-removed PUU-PCL stent had lower resistance at its smaller diameter. Statistical significance was evaluated using one-way ANOVA with Tukey's test. Data are presented as means ± SD, *$p < 0.05$.

(Fig. 4D). Compared to the commercially available stents and stents with the J-coil removed, removal of PUU-PCL stents (permanent shape, straight, smaller diameter) required significantly less force and work (Fig. 4D–F).

As demonstrated above, the ex vivo shape memory and recovery properties of PUU-PCL is desirable. Biocompatibility of PUU-PCL was then evaluated before in vivo HIFU experiments. Cell culture experiment showed that PUU-PCL is cytocompatible (Fig. S4). Degradation in PBS is slow as weight loss after 12 w incubation was not significant (Fig. S5). Subcutaneous implantation showed that PUU-PCL kept its original shape, did not induce severe inflammation reaction or significantly degrade (Fig. S6). Evaluation of the influence of aqueous extract of PUU-PCL on series of physiological functions according to ISO10993 standard showed that PUU-PCL did not induce toxic reactions (Tables S1 and S2), further demonstrating its biocompatibility.

### HIFU triggered shape recovery of PUU-PCL in canine bladder

The J shaped coil of the ureteral stent was long, and the change in coil curvature was relatively small before and after shape recovery. In addition, the canine ureter is too long and tortuous for a ureteroscope to pass though and reach the renal pelvis without injuring the ureter tissue. As a result, it is difficult to observe the shape recovery of a J shaped coil under ureteroscopy. Therefore, we chose a wrapped flag-shaped device for demonstration of shape recovery triggered by image-guided HIFU (Fig. 5A, B). HIFU triggered heating of $Fe_3O_4$ incorporated PUU-PCL in the bladder was first simulated using finite

element modeling (Fig. 5C). Simulation results showed that the acoustic pressure was precisely concentrated in a small area about 2.5 mm × 1 mm × 1 mm on the polyurethane device (Fig. 5D). The highest acoustic pressure was $4 × 10^5$ Pa. Consistent with the distribution of acoustic pressure, rapid temperature increase was observed in the same location with high acoustic pressure, and the volume of polyurethane with significant temperature increase was smaller compared to the volume of high acoustic pressure (Fig. 5E). The temperature-time curves at different probing locations in and near the energy-focused areas showed a pattern similar to the measured results in the in vitro heating experiment (Figs. 5F and 3H). In 25 s, the temperature in the center of the heated volume (P2) increased to 71 °C, the spot on the edge of the device (P1) was heated to 62 °C, and the spot on the edge of the heated volume (P3, 1 mm above the center) reached 54 °C, all above the 47 °C transition temperature of $Fe_3O_4$ nanoparticle incorporated PUU-PCL. P2 and P1 reached 47 °C in less than 5 s, significantly shorter than the time required to heat the PUU-PCL finger sample in the tissue phantom (Fig. 3H), which may be a result of less energy loss as the acoustic pressure was concentrated in the polyurethane materials of the rod-shaped device in the finite element model mimicking the wrapped flag used in vivo, and the finger device was too thin to contain the entire high acoustic pressure volume. The simulation results indicated a high heating efficiency, which is desirable for in vivo heating.

Under B mode ultrasound, the urine filled canine bladder appeared dark, and the inserted $Fe_3O_4$ incorporated PUU-PCL

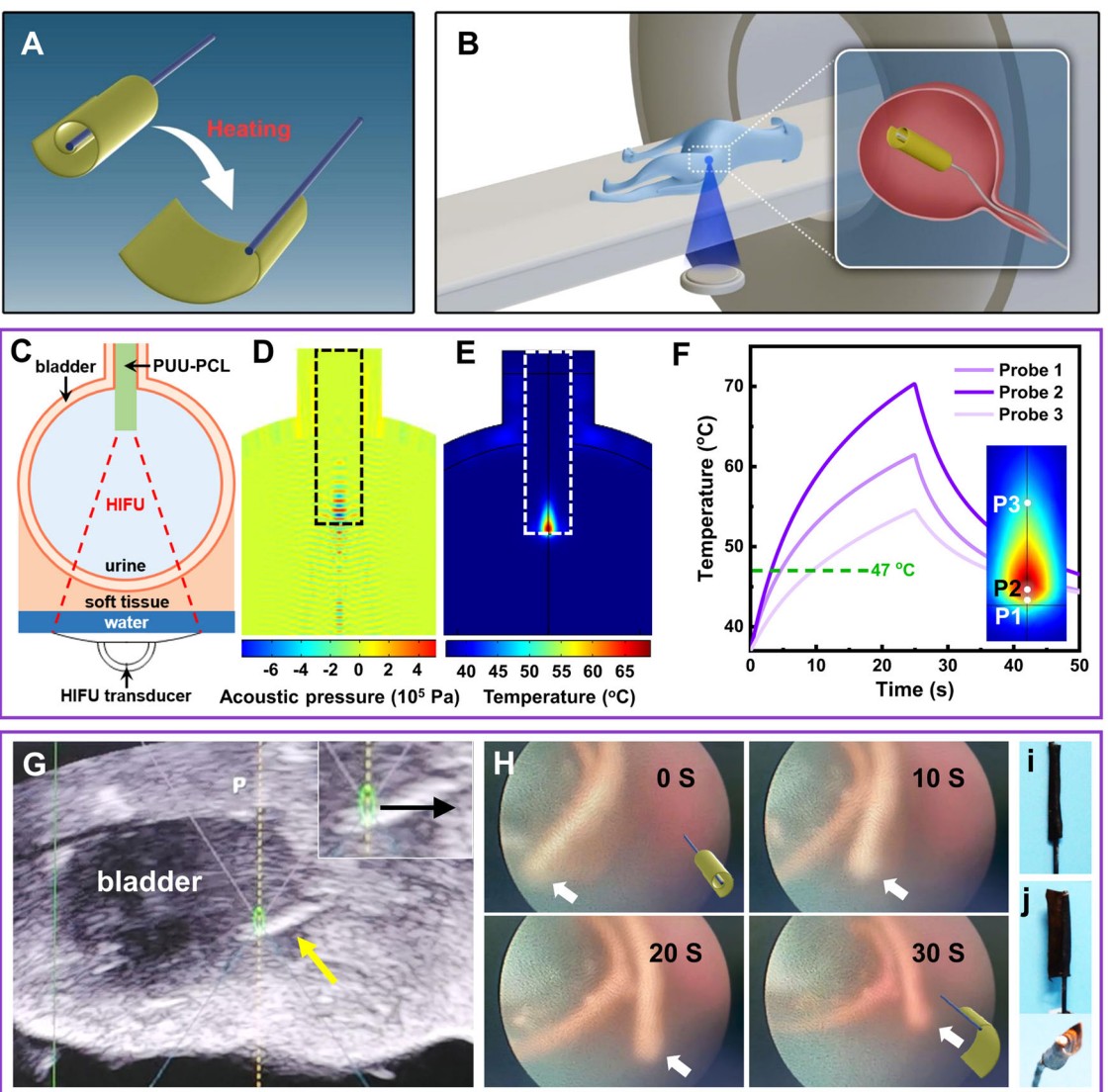

**Fig. 5 | Simulation and practice of HIFU triggered shape recovery of a PUU-PCL device in canine bladder. A** Scheme of heating induced shape recovery of the flag-shaped PUU-PCL device. **B** Set up of the HIFU heating procedure. The flag-shaped device was implanted into the canine bladder assisted via ureteroscopy. **C** HIFU energy focusing on the PUU-PCL device in canine bladder. **D** Simulated distribution of acoustic pressure on the PUU-PCL device and in the canine bladder. The black rectangle indicates the position of the PUU-PCL device. **E** Simulated temperature increase in the bladder at the end of HIFU heating period. The white rectangle shows the position of the PUU-PCL device. **F** Temperature at 3 locations in the PUU-PCL device during the HIFU heating and cooling cycle. The inset shows the positions of the 3 temperature probes. **G** B mode ultrasound image of $Fe_3O_4$

incorporated PUU-PCL device in the canine bladder. The dark area is the urine filled bladder. The wrapped PUU-PCL flag appeared bright. USgHIFU energy was focused on the implanted shape memory device, as indicated by the green ellipse. The black arrow indicates the direction in which the HIFU focus moved. **H** Shape recovery of the shape memory PUU-PCL device induced by HIFU heating in the bladder. The wrapped flag was unfolded in 30 s. The white arrows point at the movable end. The insets show the status of folded and unfolded flag device. **I** Wrapped PUU-PCL device before implantation in its temporary shape. (**J**) Unfolded device which recovered its permanent shape, and was withdrawn from the animal. Top: same view as in (**I**), bottom: front view.

appeared bright, exhibiting a high contrast for targeting (Fig. 5G). A line scan mode was adopted to cover the entire length of the wrapped flag device in 8 s. The device maintained in its original position during the heating procedure, thus absorbing all acoustic energy. The wrapped polyurethane film started to unfold approximately 2 s after heating was initiated (Fig. 5H). Unfolding continued as HIFU heating extended and the focal point moved along the device (Supplementary Movie 4). The outer layer and the inner layer of the wrapped device both unfolded as triggered by HIFU. The geometry of the withdrawn device showed more clearly the unfolding from the distal end to the proximal end of the device, compared to its temporary, wrapped state (Fig. 5I, J). In bladder, the hydraulic pressure is uniform in all directions, thus the combined force on the device before HIFU treatment is

theoretically 0 N (neglecting buoyant force). During HIFU treatment, the resistance comes from the water which needs to be pushed away by the unfolding device, which is not big to significantly hinder the movement of recovering parts of the shape memory device.

## Shape recovery triggered by image-guided HIFU is precise and safe

Finite element modeling was used to simulate HIFU induced stent heating in the renal pelvis and evaluate the risk of hyperthermia in surrounding kidney tissues. Since 1 MHz HIFU was used in the canine and sheep studies, 1 MHz HIFU was chosen in the finite element analysis to evaluate risk of potential thermal injury. The entire simulated polyurethane sample (3 mm diameter solid sphere) was heated above

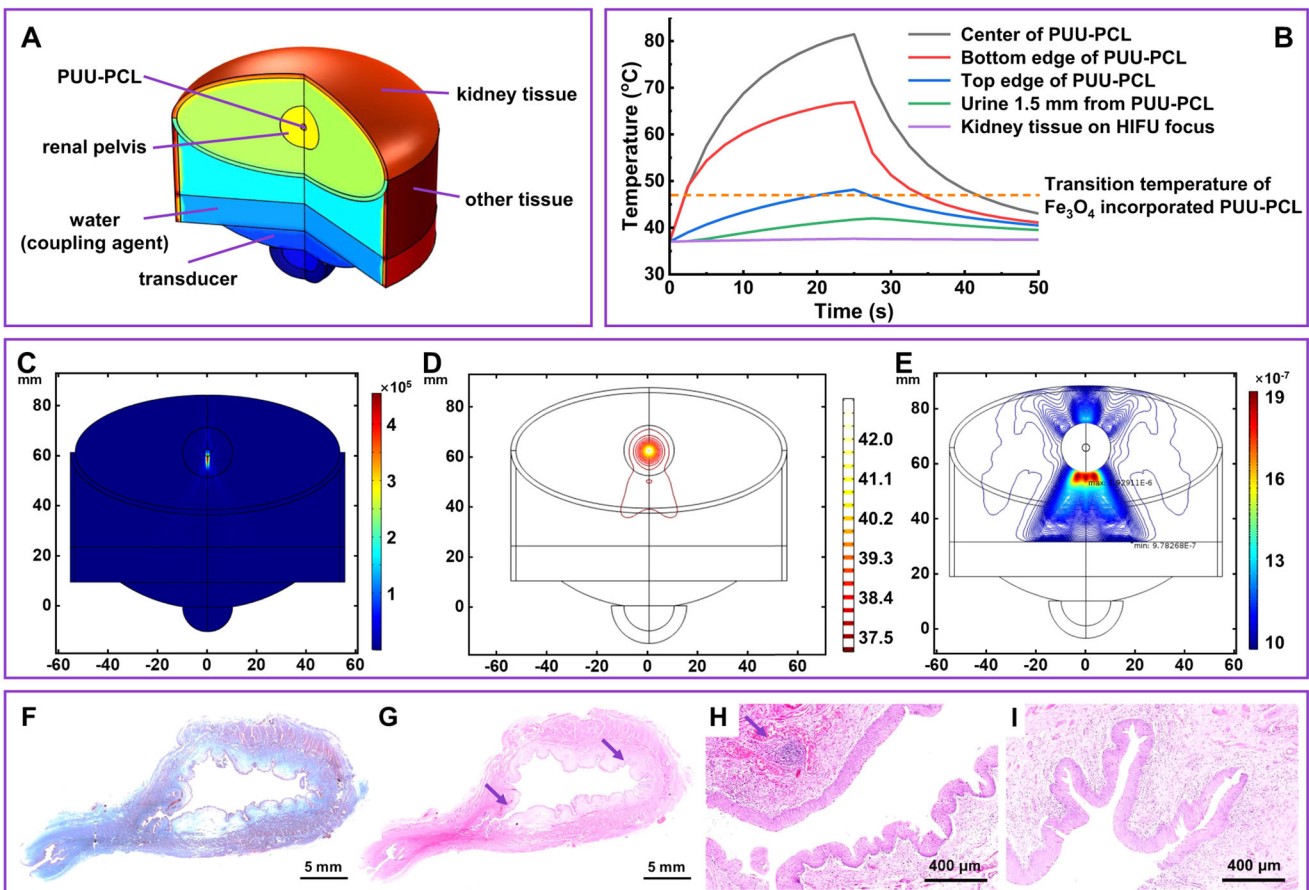

**Fig. 6 | Evaluation of hyperthermia risk in bladder of the image-guided HIFU heating procedure. A** Structural elements in the finite element model, including PUU-PCL device in the renal pelvis, surrounding kidney tissue, and coupling agent and transducer of the HIFU equipment. **B** Temperature change induced by HIFU heating in the device, adjacent urine, and kidney tissue on the boundary of renal pelvis. **C** Simulated distribution of acoustic pressure on the PUU-PCL device and in the renal pelvis. **D** Isothermal contour in the studied volume. (E) CEM43°C at different locations in the kidney and tissue underneath the kidney after one entire heating-cooling cycle. The maximum is $1.2 \times 10^{-4}$, significantly lower than the threshold of hyperthermia injury, -70. **F** Trichrome staining of the canine bladder excised 2 days post-surgery. **G** H&E staining of the same bladder in (**F**), the arrows indicate the FOI in (**H**) and (**I**). **H** H&E staining of the neck of bladder and internal urethral sphincter. The arrow indicates the micro bleeding in the muscle. (**I**) The muscle on the opposite side of the neck of bladder ($n=1$).

the transition temperature of $Fe_3O_4$ nanoparticle incorporated PUU-PCL within 25 s (at the end of heating cycle), with the peak temperature in the core reaching 81 °C, and material on the urine interface reaching 48 °C (Fig. 6A, B). HIFU induced heating was highly confined to the polyurethane sample with millimeter resolution: the peak temperature of urine 3 mm from the center of the polyurethane was 42 °C; the temperature of kidney tissue 1 mm from the renal pelvis increased by 0.6 °C (Fig. 6B). After heating stopped, PUU-PCL temperature rapidly dropped from its peak, with the dissipated heat slightly raised the temperature of the adjacent urine (<0.2 °C) before the latter quickly cooled (Fig. 6C). The temperature of each component of the system dropped below 43 °C 25 s after heating stopped (Fig. 6D). No hyperthermic injury was observed in nearby kidney tissue, the highest CEM43 °C (cumulative equivalent minutes at 43 °C, a measure for thermal dose[30]) was $1.2 \times 10^{-4}$ min (Fig. 6E), far below the injury threshold 70 min[31]. In an extreme case where HIFU was focused on the kidney tissue mimicking off-target heating (which should not happen with image-guidance), no hyperthermia injury was observed (CEM43 °C = 0.004 min).

Two days after device withdrawal, the canine bladder was excised and stained for histological evaluation of the bladder tissue. The entire mucosa, including the epithelium layer on the bladder lumen side, was intact from the bladder neck, the trigone, to the epithelium on the urachus side (Fig. 6F, G). No structural damage was found on the epithelium layer and rugae. In addition, the submucosa and the detrusor muscle were in absence of features indicative of hyperthermic injury (Fig. 6F, G). A mild enrichment of red blood cells was observed in the neck of bladder and internal urethral sphincter, indicating micro bleeding in the muscle (Fig. 6H). Since the micro bleeding was evenly distributed around the bladder neck and about 2 cm down the urethral sphincter, instead of concentrated at one spot, it was considered that the micro bleeding was secondary to the device implantation and withdrawal procedure, rather than off-target HIFU heating (Fig. 6H, I). Blood and urine tests verified the histological evaluation results. Consistent with the micro bleeding found in the sphincter, leukocyte esterase (LEU) and blood (BLD) were positive in urine 24 h and 48 h after surgery (Table S3). The color of the urine remained yellow, indicating that the bleeding was minor. White blood cell count (WBC) and total neutrophil count (NEU) slightly increased in the first 24 h and decreased back to the normal range 48 h after surgery (Table S3), indicating that the stress reaction was temporary and a result of the cystoscope and the device pressing the bladder neck, the sphincter and the urethra, rather than hyperthermic injury as the latter would be expected to continue longer. Serum creatinine, blood urea nitrogen, alkaline phosphatase, and alanine aminotransferase levels remained stable and within the normal ranges (Table S3), indicating that the HIFU triggered device shape recovery procedure did not impair kidney and liver functions. The anatomical structure and

relative positions of human, canine, and porcine bladder, ureter and kidney are similar. The length and diameter of ureters from the model animals vary from those of human. Compare to canine ureter, bladder and kidney, porcine counterparts are generally more closely resemble those of humans in size, thus porcine models could be included in future studies.

Additional large animal experiments were performed to demonstrate the potential of image-guided HIFU in triggering shape recovery of moving implants, and in triggering on-demand drug release. Black, green and yellow dye solutions were loaded in PUU-PCL tubes (one color each tube), and both ends of the tubes were folded at room temperature to seal the solution inside the tubes (forming the temporary tubular reservoir). The tubes were then fixed on the liver of a sheep, which cyclically moved with the latter at the frequency of sheep respiration. USgHIFU in scan mode precisely focused energy on the sealed ends of the dye-loaded tubes, and released all three dyes to surrounding tissue, as shown in Fig. S7.

## Discussion

As demonstrated in a large animal model, image-guided HIFU triggering of shape recovery in a PUU-PCL device could be fast and precise, able to reach a deep target and cover large volume devices. Heats of melting for typical polymers are within the range of $10^1$–$10^2$ J/g, thus the high power ($10^2$–$10^3$ W) of HIFU could provide enough energy to facilitate temperature increase to above the transition temperature in a few seconds. Results of the canine study and simulation showed that heat dissipation into surrounding body fluids and tissues can be negligible as the millimeter level precision of HIFU targeting focuses the ultrasound energy on the device. This high precision supports heating efficiency and minimizes off-target tissue thermal damage. In addition, the exhibited capacity to non-invasively heat tissue over 7 cm deep, together with reports of deeper tissue penetration including penetration through the skull[32], indicate broad accessibility of common device implant sites in the body. For shape memory devices with large size or complex geometry, HIFU supports a scan mode to traverse all locations in the device space, as employed in this study. This applies to deep, moving target devices (as shown in the experiment in which dye solution release from shape memory tubular reservoirs on moving sheep liver was triggered by HIFU). Furthermore, compared to indirect heating strategies including using coils to transfer magnetic energy to heat for triggering urethral sphincters, etc[33–36]. HIFU heating is more efficient and convenient. The above-mentioned advantages contributed to the safety and efficacy observed here for HIFU triggered device shape recovery in canine bladders. Therefore, we believe that image-guided HIFU heating has the potential to be a widely adaptable and applicable solution for triggering shape recovery of a wide variety of medical devices.

In terms of implantation sites, the renal pelvis and bladder investigated in this study, and other organs and tissues with large fluid-filled spaces (e.g. heart and large blood vessels) might be considered first for development of shape memory medical devices. Fluids encompassing the device form a physical gap between the heating target and adjacent tissue, which would allow minor faults in imaging and targeting, thus reducing the risk of focusing acoustic energy onto adjacent tissues. In addition, the high heat capacity of water renders a highly efficient heat absorbent to prevent damaging temperature increase, an effect enhanced when fluid flow is present. The mechanical resistance by body fluids to device shape change would be reduced compared to that from tissues, thus when devices are implanted in body-fluid-filled spaces, the extent and intricacy of shape change may be greater.

In addition to shape recovery, HIFU heating could trigger various device responses to incorporate other therapeutic functions. Kim et al. reported a method of triggering mechanophores by HIFU as a remote energy source to drive mechanical-to-chemical transduction of mechanoresponsive polymers with high spatial and temporal resolution[37]. Acoustic energy can be converted to light and free radical species, similar to heat, as a primary outcome or that serve as secondary triggers to initiate desired functions including biosensing, drug release, and device degradation[38,39]. Given the high spatiotemporal precision of HIFU, the locations and time of the sequence of HIFU triggered events could be programmed and orchestrated, which might allow more personalized management of disease treatment to achieve optimal therapeutic outcomes.

Despite the demonstrated safety and feasibility of HIFU triggered in vivo shape recovery of shape memory polyurethane devices, there are limitations that need to be addressed in future studies before this strategy could be applied to other shape memory biomaterials and translated to clinical application. First, the HIFU triggered shape recovery process was not connected with a specific therapeutic function. Smoother device removal for a ureteral stent may reduce morbidity associated with the procedure, but does not extend device functionality. A demonstration of shape recovery-based device implantation, drug delivery and device degradation may show higher clinical value. Secondly, the influence of HIFU heating parameters on shape recovery efficiency and precision were not investigated to reveal more general relationships between heating condition and device heating performance. For particular shape memory devices, optimization is required to achieve ideal shape recovery outcomes according to device geometry, ultrasound sensitivity, and thermal properties. In addition, a larger animal study would be needed to further consolidate the safety and efficacy of the procedure, cover different device implantation sites/heating targets and anatomic structures.

In summary, we achieved remotely triggered shape recovery of shape memory device in large animal models via image-guided HIFU heating in this study, HIFU heating of the shape memory device was fast and precise. Histological evaluation and finite element analysis demonstrated the safety of the heating procedure, in terms of a negligible risk of hyperthermia injury on surrounding tissues, attributed to the millimeter resolution. The concept of shape memory ureteral stents with lowered removal resistance upon shape recovery was proposed and demonstrated ex vivo. We believe image-guided HIFU could be a broadly applicable solution for triggering shape recovery of shape memory medical devices, and support personalized, programmed medicine on the basis of shape memory devices.

## Methods

### Research compliances
All animal experiments were approved by the Guidelines of Animal Care and Use Committees of Zhejiang University (ZJU20210167, ZJU20230348). Porcine ureters were purchased from a butcher's shop for demonstration. Except in the canine model, the sex of the animals was not considered. In canine model, female dog was chosen, mainly because the shorter urethra of the female dog can reduce the difficulty of catheter implantation and also reduce the pain of dog.

### Materials
All chemicals were obtained from Sigma-Aldrich (USA) unless otherwise stated. 1,4-diisocyanatobutane (BDI) and putrescine (1,4-diaminobutane) were purified by vacuum distillation before use. PCL diol (Mv ≈ 7200) was synthesized from ε-caprolactone and diethylene glycol.

### Synthesis of macrodiols
ε-caprolactone (CL), diethylene glycol (DEG), and δ-valerolactone (VL) (Sigma-Aldrich, USA) were purified by vacuum distillation before use. Poly(VLCL) (PVLCL) diol was synthesized with $Sn(Oct)_2$ catalyzed ring opening polymerization with DEG as an initiator, while PCL-PEG-PCL diol was synthesized with PEG as the initiator (Figure S3). Monomers and initiator were mixed and heated to 140 °C under argon protection,

to which Sn(Oct)$_2$ (0.05 wt% with respect to the monomer) was added under stirring. After 24 h reaction, macrodiols were precipitated in methanol and vacuum dried (yields of the products were >95%).

## PUU-PCL synthesis

PUU-PCL was synthesized from PCLdiol and BDI using putrescine as a chain extender by a two-step method[29] as shown in Fig. 2A, PCL: BDI: putrescine = 1: 2: 1 (molar ratio). PCL was dried by azeotropic distillation in toluene in a 3-neck flask and dissolved with anhydrous DMSO. BDI was added, followed by addition of Sn(Oct)$_2$ (0.05 wt% of PCL). The reaction was carried out for 3 h at 70 °C, and then cooled at room temperature. Putrescine/DMSO solution was added dropwise into the agitated solution. The entire PUU-PCL synthesis was protected by argon atmosphere. The product (PUU-PCL) was precipitated in deionized water and vacuum dried at 60 °C for 3 d. The yield was >95%.

## Characterization

**Chemical structure.** $^1$H-NMR spectra of PUU-PCL was recorded with a 300 MHz Bruker spectrometer using CD$_3$Cl or DMSO-d6 as a solvent. PUU-PCL films (-150 μm thick) were prepared by casting PUU-PCL solution in hexafluoroisopropanol (HFIP) (Meryer, China) onto polytetrafluoroethylene plates followed by solvent evaporation. Differential scanning calorimetry (DSC) of PUU-PCL films was conducted on a DSC-60 instrument (Shimadzu) at a rate of 10 °C/min under nitrogen flow. One dimensional x-ray diffraction (1-D XRD) for PUU-PCL were carried out on a Bruker D8 Discover XRD instrument with a CuKα source. Two dimensional XRD (2-D XRD) experiments were carried out on a Bruker Smart Apex CCD diffractometer equipped with MoKα radiation.

**Mechanical strength.** For tensile testing, dumbbell-shaped PUU-PCL samples (2.5 × 20 mm, $n = 4$) were punch cut from the PUU-PCL films and tested on an MTS Tytron 250 MicroForce Testing Workstation. Tests were carried out at room temperature in accordance with ASTM D638M-89. Both ends of the original length of L$_0$ were premarked. 30 min after sample failure, the length (L$_1$) between the two marked ends was measured and final strain was calculated as $(L_1 - L_0)/L_0 \times 100\%$. Initial modulus was calculated as the slope of the tensile curves (the initial linear region, strain <30%).

**Shape memory property.** The dumbbell-shaped PUU-PCL samples were stretched to 300% original length. The stretched samples were immersed in 20 °C water, and the water was slowly heated to 60 °C while the lengths of the samples were measured every 3 °C. The temperatures at which the samples began to shrink were recorded and considered the transition temperatures.

## In vitro PUU-PCL shape recovery induction by HIFU

A Magnetic resonance guided high-intensity focused ultrasound (MRgHIFU) system (ExAblate2000, Insightec, Haifa, Israel) with a phased array was used to induce a PUU-PCL origami. The Insightec system was equipped with a 650 kHz transducer. The table was connected to an MRI scanner (Discovery MR750w 3 T, GE, Milwaukee, WI, USA). Two standard circular Agarose gel pads as the tissue phantoms were stacked to serve as a platform for the HIFU procedure. The total thickness of these gel pads was 7 cm. The gel pads were placed on the table directly above the transducer and a 1 cm depression was cut into the exposed surface of the upper gel pad. Ultrasound gel and degassed water was used to fill this depression and the PUU-PCL samples were positioned within this ultrasound/water mixture along the exposed surface. A three-plane localizer was then performed to verify precise positioning, followed by T2-weighted treatment planning sequences. The target material shapes and surface of the upper gel pad were manually segmented (VR). HIFU heating was then prescribed on the MR planning images targeted to the folded edge of the PUU-PCL

samples. The target sample was 7 cm from the ultrasound transducer. Increases in temperature were monitored using nearly real-time MR thermometry of phase-difference fast spoiled gradient-echo sequences (proton resonant frequency shift method) in the gel pad adjacent to the target sample. MR thermometry was performed during each sonication with multiphase multislice echo planar imaging (FOV/slice thickness/TR/TE/flip angle/echo train length = 28 × 28 cm/4 mm/ 210 ms/18.3 ms/35°/12) to monitor approximate changes in temperature during the course of the procedure.

## Finite element modeling of PUU-PCL heating by HIFU in kidney and bladder

COMSOL was used to construct the human kidney geometry and simulate the HIFU heating of PUU-PCL in the renal pelvis. Figure 6 shows the geometry simulated in this model. The acoustic transducer was immersed in water. The transducer was bowl shaped with a focal length of 62.64 mm, an aperture of 35 mm in radius, and a hole of 10 mm in radius in the center. The kidney was assumed to be the shape of an ellipsoid with 55 mm long axis and 25 mm short axis, and the renal pelvis was assumed to be a spherical space with a radius of 10 mm. The PUU-PCL was simplified to be a sphere with a radius of 1.5 mm at the center of the renal pelvis, which was set as the focal point of the transducer. The soft tissue between the skin surface and the kidney was assumed to be a void free columnar tissue. The tissue and the transducer were arranged coaxially so the model could be defined as being 2-D axisymmetric. In this model, we assumed that the tissue properties did not change when the temperature rises. Blood perfusion was also neglected which could be added in subsequent work. Similarly, geometry of a canine bladder and a short PUU-PCL rod mimicking the sample (described in the in vivo HIFU experiment section) used in the in vivo HIFU triggered shape was constructed in COMSOL.

The transducer was driven at the frequency of 1 MHz as 1 MHz HIFU was used in the animal studies (Type JC USgHIFU, Haifu, China). It was turned on for 25 s and then turned off. Water mimicking the body fluids was set to be the coupling agent between the transducer and skin surface. The model used the Pressure Acoustics, Frequency Domain (ACPR) interface to model the stationary acoustic field across the tissue and PUU-PCL domain to obtain the acoustic intensity distribution in the tissue phantom. The absorbed acoustic energy was calculated and used as the heat source for the Bioheat Transfer (HT) interface model. Subsequently, the domain ordinal differential equations (Dode) module in COMSOL software was called to analyze the thermal damage field of the model.

The wave equation solved was the homogeneous Helmholtz equation in 2D axisymmetric cylindrical coordinates:

$$\frac{\partial}{\partial r}\left[-\frac{r}{\rho_c}\left(\frac{\partial p}{\partial r}\right)\right] + r\frac{\partial}{\partial z}\left[-\frac{1}{\rho_c}\left(\frac{\partial p}{\partial z}\right)\right] - \left[\left(\frac{\omega}{C_c}\right)^2\right]\frac{rp}{\rho_c} = 0 \qquad (1)$$

Here $r$ and $z$ are the radial and axial coordinates, $p$ is the acoustic pressure, and $\omega$ is the angular frequency. The density, ρc, and the speed of sound, $c_c$, are complex-valued to account for the material's damping properties. Using Eq. 1 involves the assumption that the acoustic wave propagation is linear and also that the amplitude of shear waves in the tissue domain are much smaller than that of the pressure waves. Nonlinear effects and shear waves were therefore neglected.

Given the acoustic pressure field, the acoustic intensity field was readily derived. The heat source $Q$ for thermal simulation, given in the plane-wave limit, was then calculated as:

$$Q = 2\alpha_{ABS}I = 2\alpha_{ABS}\left|\text{Re}\left(\frac{1}{2}pv\right)\right| \qquad (2)$$

where $\alpha_{ABS}$ is the acoustic absorption coefficient, $I$ is the acoustic intensity magnitude, $p$ is the acoustic pressure, and $v$ is the acoustic particle velocity vector. The heat source $Q$ was thus readily calculated once the acoustic field was solved.

Inserting the volumetric acoustic heat source into the Pennes' Bioheat Transfer equation to model heat transfer within biological tissue gives

$$\rho C_p \frac{\partial T}{\partial t} = \nabla \cdot (k\nabla T) - \rho_b C_b \omega_b (T - T_b) + Q + Q_{\text{met}} \tag{3}$$

where $T$ is the temperature, $\rho$ is the density, $C_p$ is the specific heat, k is the thermal conductivity, $\rho_b$ is the density of blood, $C_b$ is the specific heat of blood, $w_b$ is the blood perfusion rate, $T_b$ is the temperature of the blood, $Q$ is the heat source (the absorbed ultrasound energy calculated from Eq. 2), and $Q_{\text{met}}$ is the metabolic heat source.

### Ex vivo stent removal from porcine ureters
$Fe_3O_4$ nanoparticles (Sigma-Aldrich, USA) (30 wt% of PUU-PCL) were added to the PUU solution and the mixture was cast and dried on a metal wire to obtain 20 cm long PUU tubes with 2.5 mm diameter and 0.5 mm tube wall thickness. The tubes were expanded in diameter by passing a thicker metal wire through the tubes in 55 °C water and cooling the tubes back to room temperature while the metal wires were in the tubes. One end of the tubes was coiled by bending the tubes with the metal wire, generating PUU-PCL ureteral stents.

Porcine ureters were purchased from a butcher's shop and washed with PBS. The PUU-PCL ureteral stents were inserted into the porcine ureters, leaving both the coiled end and the straight end outside the corresponding ends of the porcine ureters. The assembly of one porcine ureter and one PUU-PCL stent were fixed onto a universal mechanical testing machine, with the lower clamp fixing the straight end of the PUU-PCL stent, the upper clamp fixing the porcine ureter tissue adjacent to the coiled end of the PUU-PCL stent. The PUU-PCL stents were pulled from the porcine ureters and the resistance force was recorded ($n = 3$).

### In vivo stent shape recovery induced by HIFU
PUU-PCL (1 g) was dissolved in 10 mL dioxane, followed by dispersion of 200 mg $Fe_3O_4$ nanoparticles (20-30 nm, 99.99%). The mixture was poured into a Teflon mold to obtain 2 cm × 5 cm $Fe_3O_4$ nanoparticle incorporated PUU-PCL films. The short side of the film was adhered onto a thin and long guide wire with glue, after that the film was heated (above melting temperature), and tightly wrapped around the guide wire.

This animal experiment was approved by the Guidelines of Animal Care and Use Committees of Zhejiang University (ZJU20210167). USgHIFU equipped with a 1 MHz HIFU (JC, Haifu, China) was used in the animal experiments. The dog (beagle, female, 2 years old, weighed 13.0 kg) was fasted for 8 hours before surgery to prevent reflux from blocking esophagus and trachea during anesthesia. Anesthesia of dog was induced by propofol, while heart rate and body temperature were monitored. Afterward, the anesthesia machine was connected to allow the dog to quickly enter surgical anesthesia period and fix the tracheal intubation. Multi-point subcutaneous lidocaine injection in the operation area was employed for local anesthesia.

In the animal experiment of HIFU triggered shape recovery of a PUU-PCL device in canine bladder. Ceftiofur solution was used for anti-infection. Lactated Ringer's solution was used to replenish body fluids. The dog was placed on the operation table in a prone position, and its lower abdomen was adjusted for ultrasonic probe attachment. The cystoscope was slowly pushed through the urethra into the bladder, the guide wire was introduced through the channel inside the cystoscope, followed by withdrawal of cystoscope. Under the guidance of the guide wire, the catheter could be easily inserted into the bladder

site, and the guide wire was withdrawn at the same time. The cystoscope was then reintroduced into the bladder. A guide wire with a $Fe_3O_4$ nanoparticle incorporated PUU-PCL film was introduced into the bladder through the catheter placed in the cystoscope, then the catheter was removed, and the position was adjusted until the cystoscopic view was clear. The focus point of HIFU was adjusted onto the $Fe_3O_4$ nanoparticle incorporated PUU-PCL device. The $Fe_3O_4$ nanoparticle incorporated PUU-PCL device was heated under line scan mode at a power of 420 W for 8 s from the distal end of the device to the proximal end. Shape recovery of the shape memory device was observed by cystoscopy. After shape recovery of the device, the cystoscope was retracted, first, followed by the device and guide wire. Dogs were ventilated for an extra hour before they resumed spontaneous breathing. Blood and urine were collected 24 and 48 h after surgery.

### Statistics and reproducibility
All data is provided in the manuscript. Results are shown as mean ± SD unless stated otherwise. Statistical analyses were performed by one-way ANOVA followed by Tukey's post-hoc testing. Statistical differences were considered significant when $p < 0.05$.

### Reporting summary
Further information on research design is available in the Nature Portfolio Reporting Summary linked to this article.

## Data availability
All data are available within the Main Manuscript, Supplementary Information, Source Data file, or from the authors upon reasonable request. Source data are provided with this paper.

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

## Acknowledgements

The authors thank Prof. Kevin E. Healy's assistance on characterization of PUU-PCL, Mr. Zurong Tang's assistance on HIFU experiments, Prof. Tao Xie's assistance on design of in vitro demonstration of device shape recovery, and Mr. Yang Jin's assistance on preparing the schematic figures. This study was financially supported by the National Key Research and Development Program of China (No. 2019YFE0117400, Y.Z.), National Natural Science Foundation of China (No. 82202328, Y.Z.), the Starry Night Science Fund of Zhejiang University Shanghai Institute for Advanced Study (No. SN-ZJU-SIAS-004, D.L.) and US National Institutes of Health grant R01 HL105911 (W.R.W).

## Author contributions

Y. Zhu, H. Wang, Y. Xu and W. R. Wagner designed research; Y. Zhu, K. Deng, J. Zhou, C. Lai, Z. Ma, H. Zhang, J. Pan, L. Shen, M. Bucknor, E. Ozhinsky, S. Kim, G. Chen, S. Ye, H. Wang and Y. Xu performed research; Y. Zhu, K. Deng, J. Zhou, C. Lai, Z. Ma, L. Shen, Y. Zhang, C. Gao and D. Liu analyzed data; Y. Zhu, K. Deng, Z. Ma, J. Zhou, H. Wang, Y. Xu and W. R. Wagner wrote the paper.

## Competing interests

The authors declare no competing interests.
