## [Peer Review File · Nature Communications]

REVIEWER COMMENTS

Reviewer #1 (Remarks to the Author):

In this manuscript, the authors used image-guided high-intensity focused ultrasound (HIFU) to heat the polyurethane urea unit non-invasively, accurately, and rapidly. They claimed that this is safe, efficient, and meaningful work. This work is expected to have more innovative contributions. However, as studies have already been published on shape memory urethral stents which can remotely heated and driven, compared with the published works, the advantages of this work should be much more attractive and outstanding. Therefore, it is not suitable for being published in this journal.

The comments to the manuscript are as follows:

1. The clarity of Figure 2 could be better, especially the text in Figure 2c, which is difficult to read. There is no Fig. 2 (G) related description in the caption of Fig. 2.
2. The working process of ureteral stents is based on the shape recovery performance of shape memory materials. Still, the performance characterization of PUU-PCL materials in this paper needs to be improved, especially the characterization of shape memory performance, including shape fixation and recovery rates.
3. It is well known that the shape-restoring force of shape memory polymers is very small. The fluid existing in the bladder hinders the shape recovery of the stent, especially for the flag-shaped structure which may subject to hydraulic resistance. Is there any method used to guarantee the shape recovery of the flag-shaped device in the fluidic environment?
4. According your design, how long will the ureteral stent be placed in the body? How about the biocompatible properties of the stent or materials? If it should work for a longer time, how about its biodegradability?
5. The biocompatibility of biomaterials is one of the critical indicators, and it is suggested that the author supplement the compatibility experiment of this PUU-PCL.
6. It is suggested that the author describes in detail the molecular mechanism of PUU-PCL under ultrasound and how to achieve the shape recovery of the material under ultrasound. At present, the mechanism needs to be more profound and more comprehensive.

Reviewer #2 (Remarks to the Author):

Very interesting technology and testing.

Using the canine ureter can be difficult.

the videos are a little difficult to visualize and would recommend just doing the video in a test tube to demonstrate how they react to temperature changes.

The changes in forces required to remove the stent are nicely done.

How biocompatible is the material? Has pathology on the ureters been performed to test the biocompatibility? I don't know if you need it for this manuscript, but I think this is something for another study to ensure the material does not elicit an inflammatory reaction. I would venture to say that this (the type of material and how biocompatible it is) is more of a problem than the coil being dragged out during removal which your technology tends to address.

But very interesting data. Pigs could also be used as well too since their ureters most closely resemble those of humans in size.

Reviewer #3 (Remarks to the Author):

Achieving remotely controlled shape recovery of shape memory instrumentation by HIFU will be revolutionary in applications where medical devices need to be implanted or removed. The authors demonstrate this in simple experiments. The study is overall clearly outlined.

Please replace citation 18 with the citation below. This is the pioneer study at NEJM that led to the FDA approval of HIFU application in the brain (please note I have no affiliation with the paper NOR am I a co-author of this paper): A Randomized Trial of Focused Ultrasound Thalamotomy for Essential Tremor.

Strictly speaking, image-guided MR thermometry is not precisely real-time. There is an offset of a few minutes depending on the MR sequence. Please replace this section in the paper by indicating that this procedure is nearly real-time or similar.

When reporting the MR sequence details, please also write the corresponding SI units.

In the finite element model, the authors apply an ultrasound frequency of 1 MHz. The InSightec system operates at a center frequency of 650 kHz. Please explain why 1 MHz was chosen over 650 kHz.

Please describe in detail the HIFU settings that were prescribed including the length (e.g., pulse length, duration of prescription, duty cycle, energy).

For statistics, please indicate in the bar plot figure legend what the star corresponds to (e.g., 0.05%, etc). The statistical analysis section is very scarce. Please elaborate more. If there is no statistician on the team, please include a statistician who can help.

HIFU is not mechanical but thermal in nature. Please remove citation 14.

In general, citations in the manuscript are very scarce. The authors should revise the citations and include more citations from original research rather than review articles.

Response to reviewers

Reviewer 1:

In this manuscript, the authors used image-guided high-intensity focused ultrasound (HIFU) to heat the polyurethane urea unit non-invasively, accurately, and rapidly. They claimed that this is safe, efficient, and meaningful work. This work is expected to have more innovative contributions. However, as studies have already been published on shape memory urethral stents which can remotely heated and driven, compared with the published works, the advantages of this work should be much more attractive and outstanding. Therefore, it is not suitable for being published in this journal. The comments to the manuscript are as follows:

Answer: We greatly appreciate the reviewer's comments and suggestions. The key innovation of this study is in the HIFU based remote, non-invasive, and precise heating strategy of implanted shape memory medical devices. The shape memory ureteral stent served as an example medical device/implant.

There are shape memory ureteral stents, and their deformation in vivo is mostly triggered by injecting warm and cold saline into the urethra. We searched the literature and failed to find records of deformation of shape memory ureteral stents being remotely triggered. We did find manuscripts describing artificial sphincter systems which could be triggered via integrated transcutaneous energy transfer systems, which include coils receiving magnetic energy to either heat shape memory material in the sphincters, or charge batteries powering the actuators on the sphincters. These remote trigger designs are representative in terms of integration of specific parts in the implants to receive energy remotely. In addition to the extra volume and complexity, the efficiency of power transmission rapidly decreases as the distance between the in vivo coil and external coil

increases, which inhibits the application of the triggering strategy on other shape memory medical devices, particularly these implanted deep in the body. Furthermore, the coil based energy transfer strategy is an indirect method to heat the shape memory device.

Comparison with state-of-art technologies

	Photothermal Chu et al.	Induction heating Müller et al.	HIFU Our work
model	rat, open chest	rat, noninvasive	canine, noninvasive
depth	~2 mm	~5 mm	>5 cm
target resolution	mm in open surgery; theoretically cm in closed surgery	entire device	mm
speed	60 s 0.4 mm long device	>20 s 10 mm long device	8 s 20 mm long device
power requirement	N.A. in noninvasive surgery	10 ⁴ W in rat	10 ² W in canine
equipment	N.A. for noninvasive surgery in human	N.A. for noninvasive surgery in human	clinically available, FDA approved for tumor ablation

A universal remote heating strategy which is suitable for a wider spectrum of shape memory devices and corresponding application scenarios, and does not require specific accessories to receive triggering energy, is more favorable. As summarized below, previous in vivo studies

include remote heating with light and magnetic fields. Photo energy dissipates rapidly as it penetrates soft tissue, thus it can hardly be focused on deep targets. As a result, photothermal heating of shape memory devices has only been achieved in a rat open wound surgery. Induction heating by alternating magnetic field theoretically requires residing the human or animal body in a coil. This has only been used in small animal models, and the risks of “microwaving” soft tissue in the magnetic field have not been thoroughly evaluated. In addition to the safety and efficacy in large animal model, our results demonstrated the advantages in depth, resolution, speed, and maturity in technology.

In addition to the aforementioned advantages, we performed additional large animal experiments to demonstrate the potential of image-guided HIFU in triggering shape recovery of deeply implanted moving devices, and in triggering on-demand drug release, which is difficult to achieve by other strategies. Dye solution was loaded in a PUU-PCL tube, and both ends of the tubes were folded at room temperature to seal the solution inside the tubes (forming the temporary tube shape). The tubes were then fixed on the liver of a sheep, which cyclically moved with the latter. USgHIFU precisely focused energy on the sealed ends of the dye-loaded tubes, and released the dyes to surrounding tissue. As shown in **Figure R1** (added as **Figure S7**), black, green, and yellow dyes were loaded separately in 3 tubes, line scan HIFU successfully triggered dye release from all three tubes which were constantly moving with the liver at the frequency of sheep respiration.

The following text was added to the Discussion section to further elaborate the advantages of HIFU-based remote, noninvasive triggering strategy for shape recovery of medical devices, including deeply implanted moving devices.

“...For shape memory devices with large size or complex geometry, HIFU supports a scan mode to traverse all locations in the device space, as employed in this study. This applies to deep, moving target devices (as shown in the experiment in which dye solution release from shape memory tubular reservoirs on moving sheep liver was triggered by HIFU). Furthermore, compared to indirect heating strategies including using coils to transfer magnetic energy to heat for triggering urethral sphincters, etc³³⁻³⁶, HIFU heating is more efficient and convenient....”

HIFU triggered shape recovery of **moving** devices and dye release on sheep liver

Figure R1. USgHIFU triggered in vivo shape recovery of deeply implanted moving devices and dye solution release. (A) Scheme of animal experiment including patch implantation, HIFU triggered shape recovery, and dye release. (B) B mode ultrasound image of PUU-PCL tubes (loaded with Fe_3O_4 nanoparticles, fixed on a PDMS patch) on moving sheep liver. The plane of PUU-PCL tubes is indicated by the green line. (C) Sealed PUU-PCL tubes loaded with dye solution (black, green, and yellow dye in different tubes). Yellow arrows: tube ends sealed at room temperature. (D) Dye solution loaded tubes implanted on a sheep liver. The tubes moved with the liver at the frequency of sheep respiration. (E) USgHIFU treated tubes. The sealed ends reopened,

as pointed by the white arrows. All three dyes released from the tubes can be observed. Scale bar = 1 cm.

A paragraph describing the results of the dye release experiment was added at the end of the Results section.

*“Additional large animal experiments were performed to demonstrate the potential of image-guided HIFU in triggering shape recovery of moving implants, and in triggering on-demand drug release. Black, green and yellow dye solutions were loaded in PUU-PCL tubes (one color each tube), and both ends of the tubes were folded at room temperature to seal the solution inside the tubes (forming the temporary tubular reservoir). The tubes were then fixed on the liver of a sheep, which cyclically moved with the latter at the frequency of sheep respiration. USgHIFU in scan mode precisely focused energy on the sealed ends of the dye-loaded tubes, and released all three dyes to surrounding tissue, as shown in **Figure S7.**”*

Figure R1 was added as Figure S7. Corresponding description of experiment methods were added to SI as well.

“USgHIFU triggered shape recovery of moving devices and dye release on sheep liver

In the animal experiment of HIFU triggered shape recovery of a PUU-PCL device on sheep liver. Ceftriaxone solution was used for anti-infection. Lactated Ringer’s solution was used to replenish body fluids. The above patch integrated with PUU-PCL tubes was fixed on the liver of a sheep one day before HIFU operation, monitoring sheep physical condition and indicators. The surgery was performed next day. The anesthetized sheep was placed on the operation table in a lateral position and the body was adjusted for ultrasonic probe attachment. The focus point of HIFU was adjusted onto the Fe₃O₄ nanoparticle incorporated PUU-PCL tubes. The Fe₃O₄ nanoparticle incorporated PUU-PCL tubes were heated under line scan mode at a power of 400

W for 10 s from the distal end of the device to the proximal end. After HIFU heating, the patch was taken out from sheep liver for observation. Sheep was ventilated for an extra hour before they resumed spontaneous breathing. The HIFU parameters used in the animal studies are listed in Table S4.”

1. The clarity of Figure 2 could be better, especially the text in Figure 2c, which is difficult to read. There is no Fig. 2 (G) related description in the caption of Fig. 2.

Answer: We have modified Figure 2 for higher quality, as pasted below.

2. The working process of ureteral stents is based on the shape recovery performance of shape memory materials. Still, the performance characterization of PUU-PCL materials in this paper needs to be improved, especially the characterization of shape memory performance, including shape fixation and recovery rates.

Answer: We agree with the reviewer that the shape memory performance of PUU-PCL is important. In Figure 2, we characterized the phase transition temperature and crystallinity changes

during phase changes. Between shape fixation rate and shape recovery rate, the latter is more critical in our opinion, as in our design, the PUU-PCL medical devices are implanted in their temporary shapes. This is also how most shape recovery devices are used (implant in temporary shapes, or implant in permanent shapes and then change to temporary shape in vivo). Shape recovery rates of PUU-PCL ureteral stents and tubes in water were evaluated, as shown in **Figure R2**. The PUU-PCL devices returned to their permanent shapes in less than 3 s. To better mimic the focused heating as HIFU does, we heated the folded ends of a PUU-PCL tube (similar to the ones used in the sheep study) with laser. The tube end recovered its original shape within 10 s, as shown in **Supplementary Movie 3**. The heating rate of PUU-PCL under HIFU treatment was measured, as shown in **Figure 3**. At 150 W heating, MRgHIFU heated PUU-PCL from room temperature to its transition temperature (53°C) in about 30 s. Under the same condition, it took about 25 s from 37°C to 53°C , and about 40 s to cool back down to 37°C . In vivo, USgHIFU started to unfold the wrapped flag device within 2 s of treatment, the entire flag device started to unwrap in 8 s, and the device completely unfolded in 20 s. The in vivo experiments showed that the response of PUU-PCL to HIFU heating is desirably fast.

Figure R2. Shape recovery rates of PUU-PCL ureteral stents and tubes in vitro. (A) Temperature triggered shape recovery of single-J stents in vitro, which was recovered within 2.8 s. (B) Temperature triggered shape recovery of PUU-PCL tube in vitro within 2.4 s at 55 °C, which leads to dye solution release from the tube (white arrow). This did not occur at 25 °C. (C) 808 nm laser triggered shape recovery of PUU-PCL tube in vitro. (i) The initial shape of tube. Both ends were folded. (ii) 808 nm laser triggered recovery of end 1 within 6 s and end 2 within 10 s. (iv) The final shape of tube with opened ends.

Text in the Results section have been modified accordingly:

“...The nanoparticle-incorporated tubes were coiled into single-J stents and expanded to match their diameter to commercial products, as shown in **Figure 4A**. Stents in the temporary shape could swiftly recover to the permanent shape (uncoil of the single-J part and decrease in diameter

within 3 s, as shown in Supplementary Movie 1). Fast recovery was also achieved in other tests. Sealed tubes reopened and released the loaded dye upon heating in 55°C water in less than 3 s, as well (Supplementary Movie 2). Laser targeted heating recovered the permanent shape of a PUU-PCL tube (visualized mimicking of HIFU heating) in < 10 s (for each folded end, as shown in Supplementary Movie 3). In an ex vivo stent removal assessment, stents were pulled through a porcine ureter (Figure 4B)....”

Shape fixation of PUU-PCL devices are also fast. When stretched at room temperature, the PUU-PCL samples fixed their shapes in less than 1 s, attributed to the fast crystallization of the PCL segments. When deformed above PCL melting temperature, the resulted temporary shapes could also be fixed instantly by cooling with cold water or room temperature water.

“The PUU-PCL samples can be stretched to >500% strain, and exhibited classical cold drawing phenomenon during stretching (Figure 2D, Figure S2). The temporary shapes of PUU-PCL devices were fixed instantly (< 1 s) after being deformed at room temperature, or cooled in cold water after deformed above PCL melting temperature. The stretch induced crystalline domains can be melted in 53°C saline, resulting in instant shape recovery of the samples...”

3. It is well known that the shape-restoring force of shape memory polymers is very small. The fluid existing in the bladder hinders the shape recovery of the stent, especially for the flag-shaped structure which may subject to hydraulic resistance. Is there any method used to guarantee the shape recovery of the flag-shaped device in the fluidic environment?

Answer: We thank the reviewer for the question. In bladder, the hydraulic pressure is uniform in all directions, thus the combined force on the device before HIFU treatment is theoretically 0 N (neglecting buoyant force). During HIFU treatment, the resistance comes from the water which needs to be pushed away by the unfolding device, which is not large enough to significantly hinder

the movement of recovering parts of the shape memory device, as demonstrated in our animal study, and in vitro experiments showed in **Figure R2, Supplementary Movie 1, and Supplementary Movie 1**.

We found an interesting study in the literature, in which a soft robot swam in deep sea (**Figure R3**). This work supports the analysis above: hydraulic pressure does not inhibit the movement of polymer devices/parts, as long as it does not crash the device and is uniform in all directions (neglecting buoyant force).

Figure R3. *Front (A) and top views (B) of the soft robot swimming at 3,224 m depth. Figure adapted from Li et al., Nature 2021.*

Text in the Results section of the manuscript at have been modified accordingly:

“The geometry of the withdrawn device showed more clearly the unfolding from the distal end to the proximal end of the device, compared to its temporary, wrapped state (Figure 5I,J). In bladder, the hydraulic pressure is uniform in all directions, thus the combined force on the device before HIFU treatment is theoretically 0 N (neglecting buoyant force). During HIFU treatment, the resistance comes from the water which needs to be pushed away by the unfolding device, which is not big to significantly hinder the movement of recovering parts of the shape memory device.”

4. According your design, how long will the ureteral stent be placed in the body? How about the biocompatible properties of the stent or materials? If it should work for a longer time, how about its biodegradability?

Answer: Ureteral stents are usually placed in the body for weeks to months, according to patient condition. Shape memory ureteral stents theoretically stay as long as typical stents.

To evaluate the biocompatibility of PUU-PCL, we cultured smooth muscle cells on the surface of PUU-PCL films. After 1 or 7 days of culture on the films, live/dead staining and MTS assay were employed. Live/dead staining showed that compared to tissue culture plates (TCPS), PUU-PCL did not significantly increase the percentage of dead cells (**Figure R4A**). This result agrees with MTS assay results, which did not show significant differences between PUU-PCL group and TCPS group (**Figure R4B**). These results showed that PUU-PCL material is cytocompatible.

Figure R4 was added to supporting information as **Figure S4**.

PUU-PCL degrades slowly in PBS. The weight loss of PUU-PCL films is negligible after being incubated in PBS for 12 weeks, as shown in **Figure R5**. **Figure R5** was added to supporting information as **Figure S5**.

Figure R4. *Cytocompatibility of PUU-PCL. (A) Live/dead staining. (B) MTS assay (n = 3).*

Figure R5. Degradation of PUU-PCL films in PBS ($n = 3$).

Text in the Results section was modified accordingly:

“As demonstrated above, the ex vivo shape memory and recovery properties of PUU-PCL is desirable. Biocompatibility of PUU-PCL was then evaluated before in vivo HIFU experiments. Cell culture experiment showed that PUU-PCL is cytocompatible (Figure S4). Degradation in PBS is slow as weight loss after 12 w incubation was not significant (Figure S5). Subcutaneous implantation showed that PUU-PCL kept its original shape, did not induce severe inflammation reaction or significantly degrade (Figure S6). Evaluation of the influence of aqueous extract of PUU-PCL on series of physiological functions according to ISO10993 standard showed that PUU-PCL did not induce toxic reactions (Table S1 and S2), further demonstrating its biocompatibility.”

- 5. The biocompatibility of biomaterials is one of the critical indicators, and it is suggested that the author supplement the compatibility experiment of this PUU-PCL.**

Answer: We agree with the reviewer that biocompatibility of PUU-PCL is a critical indicator. We evaluated PUU-PCL and Fe₃O₄ incorporated PUU-PCL with a rat subcutaneous implantation experiment, as a commercially available stent served as the control group. Four weeks after subcutaneous implantation of the stent samples (short tubes as part of the stents), rat blood was collected for evaluation of hepatorenal safety. Then the subcutaneous tissue around the implanted tube and vital organs, including the liver and kidney, were harvested for histopathologic examination. As shown in **Figure R6**, indicators of hepatorenal toxicity in the blood including AST, ALT, CRE and UREA showed no significant difference between PUU-PCL groups and the control. Consistently, no significant pathological changes were found in the organs of all groups, demonstrating the biocompatibility of PUU-PCL and Fe₃O₄ incorporated PUU-PCL.

Acute systemic toxicity was evaluated by Hangzhou Tigermed Testing, a third-party institution possesses China National Accreditation Service (CNAS) qualification and China Metrology Accreditation (CMA) qualification, issued by CNAS/CMA certified laboratory test report (ISO/IEC 17025). The test was performed according to the ISO 10993.11-2017 standards for Biological evaluation of medical devices - Part 11: Tests for systemic toxicity. The test article was extracted in 0.9% sodium chloride (SC) and cottonseed oil (CSO), the control vehicle would be the same method as test extract but without test article. Each polar extract was intravenously injected to five ICR mice. Each non-polar extract was intraperitoneally injected to five test mice. The general state, toxicity and number of dead animals were observed at 4h, 24h, 48h and 72h after injection, and the animal weight was recorded daily. Clinic pathological and gross pathology evaluations were performed. The clinical symptoms to be observed were listed in **Table R1**. If any gross pathology abnormality is observed at autopsy, histopathological examination is made. Body weight of all the experimental animals increased obviously, and there was no obvious abnormality

in diet and growth state. Clinical observations and body weights data are shown in **Table R2**. Clinical pathology and gross pathology evaluation: Not performed as there were no dead animals and no clinical symptoms during the experiment. Under the conditions of this test, the SC and CSO extracts of the test article showed no evidence of causing acute systemic toxicity in mice.

Figure R6. (A) Hepatorenal safety of implanted tubes by blood detection ($n = 4$). Statistical significance was calculated using one-way ANOVA with Tukey's test, and data are presented as means \pm SEM. $ns > 0.05$. (B) Histopathologic examination of subcutaneous tissue around the implanted tube with larger magnification insets (representative for $n = 4$ biologically independent

samples). Scale bar (blue) = 2 mm. Scale bar (black) = 200 μ m. (C) Histopathologic examination of vital organs (representative for n = 4 biologically independent samples). Scale bar = 200 μ m.

Table R1. Observation of toxic reactions

Clinical observation	Observed sign	Involved system(s)
Respiratory	Dyspnea (abdominal breathing, gasping), apnoea, cyanosis, tachypnea, nostril discharges	CNS, pulmonary, cardiac
Motor activities	Decrease/increase somnolence, loss of righting, catalepsy, ataxia, unusual locomotion, prostration, tremors, fasciculation	CNS, somatomotor, sensory, neuromuscular, autonomic
Convulsion	Clonic, tonic, tonic-clonic, asphyxial, opisthotonos	CNS, neuromuscular, autonomic, respiratory
Reflexes	Corneal, righting, myotact, light, startle reflex	CNS, sensory, autonomic, neuromuscular,
Ocular signs	Lacrimation, miosis, mydriasis, exophthalmos, ptosis, opacity, iritis, conjunctivitis, chromodacryorrhea, relaxation of nictitating membrane	Autonomic, irritation
Cardiovascular signs	Bradycardia, tachycardia, arrhythmia, vasodilation, vasoconstriction,	CNS, autonomic, cardiac, pulmonary
Salivation	Excessive	Autonomic
Piloerection	Rough hair	Autonomic
Analgesia	Decrease reaction	CNS, sensory
Muscle tone	Hypotonia, hypertonia	Autonomic
Gastrointestinal	Soft stool, diarrhoea, emesis, diuresis, rhinorrhea	CNS, autonomic, sensory, GI motility, kidney
Skin	Oedema, erythema	Tissue damage, irritation

Table R2. Body weight at different times and clinical observation results of test animals after treatment

Group	Treatment pathway	Animals number	Sex	Weight(g)				Clinical observation				
				Initial	24h	48h	72h	Immediate	4h	24h	48h	72h
Test group	IV.	231083	Female	18.5	19.4	20.4	21.2	—	—	—	—	—
		231097		19.2	19.8	21.0	22.0	—	—	—	—	—
		231073		19.6	21.3	23.0	24.4	—	—	—	—	—
		231096		18.7	19.5	20.5	21.4	—	—	—	—	—
		231100		18.2	18.6	19.6	20.9	—	—	—	—	—
	IP.	231072	Female	18.7	19.8	20.2	21.1	—	—	—	—	—
		231082		18.3	19.3	20.0	20.7	—	—	—	—	—
		231079		19.9	20.2	20.9	22.0	—	—	—	—	—
		231102		19.0	19.7	20.7	21.6	—	—	—	—	—
		231088		20.8	21.1	22.3	23.2	—	—	—	—	—
Control group	IV.	231084	Female	18.3	19.6	20.4	21.9	—	—	—	—	—
		231074		18.7	19.4	20.4	21.7	—	—	—	—	—
		231103		20.4	21.6	22.6	24.2	—	—	—	—	—
		231086		20.9	21.8	22.7	23.9	—	—	—	—	—
		231104		19.9	21.3	22.1	23.5	—	—	—	—	—
	IP.	231071	Female	19.3	20.1	20.7	22.1	—	—	—	—	—
		231095		20.6	21.5	22.7	23.6	—	—	—	—	—
		231085		18.4	20.0	20.8	21.5	—	—	—	—	—
		231093		19.8	20.4	21.4	22.3	—	—	—	—	—
		231105		21.8	23.1	24.3	25.1	—	—	—	—	—

IV. intravenous injection. IP. intraperitoneal injection. “ - ” : the animal is normal after injection.

Text in the Results section of the manuscript was modified accordingly:

“As demonstrated above, the ex vivo shape memory and recovery properties of PUU-PCL is desirable. Biocompatibility of PUU-PCL was then evaluated before in vivo HIFU experiments. Cell culture experiment showed that PUU-PCL is cytocompatible (Figure S4). Degradation in

PBS is slow as weight loss after 12 w incubation was not significant (Figure S5). Subcutaneous implantation showed that PUU-PCL kept its original shape, did not induce a severe inflammatory reaction or significantly degrade (Figure S6). Evaluation of the influence of aqueous extract of PUU-PCL on series of physiological functions according to ISO10993 standard showed that PUU-PCL did not induce toxic reactions (Table S1 and S2), further demonstrating its biocompatibility.”

6. It is suggested that the author describes in detail the molecular mechanism of PUU-PCL under ultrasound and how to achieve the shape recovery of the material under ultrasound. At present, the mechanism needs to be more profound and more comprehensive.

Answer: We thank the reviewer for the question. Similar to many other PCL-based shape memory polyurethanes, and as described in **Figure 2G**, when being stretched, PCL segments crystallizes and fix the temporary shape. **Figure 2B** and **Figure 2C** demonstrated the increased crystallinity of PCL in PUU-PCL. While the temporary shape is fixed, the network of the stretched polymer chains stores the energy. Under ultrasound, the PUU-PCL sample and water surrounding the sample are heated by ultrasound driven high-frequency vibration, which melt the microdomains of crystallized PCL, as shown by the DSC results in **Figure 2F**. The energy stored in the stretched PUU-PCL network releases, and the network shrinks back to its original size, as macroscopically demonstrated by sample length measurement shown in **Figure 2D**. As the polymer network shrinks, the PUU-PCL devices recover their original shapes. This process is also facilitated by increased polymer chain mobility at high temperature (above the melting point of PCL). The mechanism is summarized in **Figure 2G**, which has been reported in many other PCL based shape memory polymers¹⁻⁵

The related text in the Result section is copied below:

“The structure of a PUU with poly(caprolactone) soft segment (PUU-PCL) (Figure 2A) was confirmed by $^1\text{H-NMR}$ as shown in Figure S1. ^1H peaks characteristic of the soft segments and characteristic peaks from urea and urethane groups of the hard segment can be found in Figure S1. Stretch-induced molecular orientation and re-crystallization of PUU-PCL was investigated by wide angle XRD. In the 2-D XRD patterns of the non-stretched PUU-PCL, (Figure 2B), a broad circle corresponding to 110 and 200 lattice planes of PCL can be observed. In the stretched PUU-PCL, the circle pattern transferred into discontinuous, symmetrical arcs, indicating stretch-induced molecular orientation and re-crystallization²⁹. On the 1-D XRD spectrum (CuK α) for the non-stretched sample, peaks at $2\theta=21.6^\circ$ and 24° were observed, which were consistent with the diffraction of the 110 and 200 lattice planes of orthorhombic crystalline PCL (Figure 2C). In the stretched samples, the peaks corresponding to 90° to the stretched direction become significantly stronger as a result of stretch enhanced crystallization. In contrast, the peaks of the stretched direction were weaker compared to the ones of the non-stretched samples.”

Reviewer 2:

Very interesting technology and testing. Using the canine ureter can be difficult. The videos are a little difficult to visualize and would recommend just doing the video in a test tube to demonstrate how they react to temperature changes. The changes in forces required to remove the stent are nicely done. How biocompatible is the material? Has pathology on the ureters been performed to test the biocompatibility? I don't know if you need it for this manuscript, but I think this is something for another study to ensure the material does not elicit an inflammatory reaction. I would venture to say that this (the type of material and how biocompatible it is) is more of a problem than the coil being dragged out during removal which your technology tends to address. But very interesting data. Pigs could also be used as well too since their ureters most closely resemble those of humans in size.

Answer: We greatly appreciate the reviewer's acknowledgement on our work.

1. The videos are a little difficult to visualize and would recommend just doing the video in a test tube to demonstrate how they react to temperature changes.

Answer: We thank the reviewer's suggestion. In addition to the video of PUU-PCL ureteral stent recovering in a graduated cylinder, we recorded two other videos to demonstrate how PUU-PCL devices react to temperature changes. (1) Sealed ends of a PUU-PCL tube reopen in warm water and releases loaded dye (*Supplementary Movie 2, Figure 2B*). (2) Sealed ends of a PUU-PCL tube remote reopen triggered by the 808 nm laser (*Supplementary Movie 3, Figure 2C*).

Figure R2. Shape recovery rates of PUU-PCL ureteral stents and tubes in vitro. (A) Temperature triggered shape recovery of single-J stents in vitro, which was recovered within 2.8 s. (B) Temperature triggered shape recovery of PUU-PCL tube in vitro within 2.4 s at 55 °C, which leads to dye solution release from the tube (white arrow). This did not occur at 25 °C. (C) 808 nm laser triggered shape recovery of PUU-PCL tube in vitro. (i) The initial shape of tube. Both ends were folded. (ii) 808 nm laser triggered recovery of end 1 within 6 s and end 2 within 10 s. (iv) The final shape of tube with opened ends.

2. How biocompatible is the material? Has pathology on the ureters been performed to test the biocompatibility?

Answer: To evaluate the biocompatibility of PUU-PCL, we performed additional in vitro and in vivo experiments. We cultured smooth muscle cells on the surface of PUU-PCL films. After 1 or

7 days of culture on the films, live/dead staining and MTS assay were employed. Live/dead staining showed that compared to TCPS, PUU-PCL did not significantly increase the percentage of dead cells (**Figure R4A**). This result agrees with MTS assay results, which did not show significant differences between PUU-PCL group and TCPS group (**Figure R4B**). These results showed that PUU-PCL material is cytocompatible.

Figure R4. *Cytocompatibility of PUU-PCL. (A) Live/dead staining. (B) MTS assay.*

We evaluated PUU-PCL and Fe₃O₄ incorporated PUU-PCL with a rat subcutaneous implantation experiment, as a commercially available stent served as the control group. Four weeks after subcutaneous implantation of the stent samples (short tubes as part of the stents), rat blood was collected for evaluation of hepatorenal safety. Then the subcutaneous tissue around the implanted tube and vital organs, including the liver and kidney, were harvested for histopathologic examination. As shown in **Figure R6**, indicators of hepatorenal toxicity in the blood including AST, ALT, CRE and UREA showed no significant difference between PUU-PCL groups and the control. Consistently, no significant pathological changes were found in the organs of all groups, demonstrating the biocompatibility of PUU-PCL and Fe₃O₄ incorporated PUU-PCL.

Acute systemic toxicity was evaluated by Hangzhou Tigermed Testing, a third-party institution possesses China National Accreditation Service (CNAS) qualification and China Metrology Accreditation (CMA) qualification, issued by CNAS/CMA certified laboratory test report (ISO/IEC 17025). The test was performed according to the ISO 10993.11-2017 standards for Biological evaluation of medical devices - Part 11: Tests for systemic toxicity. The test article was extracted in 0.9% sodium chloride (SC) and cottonseed oil (CSO), the control vehicle would be the same method as test extract but without test article. Each polar extract was intravenously injected to five ICR mice. Each non-polar extract was intraperitoneally injected to five test mice. The general state, toxicity and number of dead animals were observed at 4h, 24h, 48h and 72h after injection, and the animal weight was recorded daily. Clinic pathological and gross pathology evaluations were performed. The clinical symptoms to be observed were listed in **Table R1**. If any gross pathology abnormality is observed at autopsy, histopathological examination is made. Body weight of all the experimental animals increased obviously, and there was no obvious abnormality in diet and growth state. Clinical observations and body weights data are shown in **Table R2**. Clinical pathology and gross pathology evaluation: Not performed as there were no dead animals and no clinical symptoms during the experiment. Under the conditions of this test, the SC and CSO extracts of the test article showed no evidence of causing acute systemic toxicity in mice.

Figure R6. (A) Hepatorenal safety of implanted tubes by blood detection ($n = 4$). Statistical significance was calculated using one-way ANOVA with Tukey's test, and data are presented as means \pm SEM. $ns > 0.05$. (B) Histopathologic examination of subcutaneous tissue around the implanted tube with larger magnification insets (representative for $n = 4$ biologically independent samples). Scale bar (blue) = 2 mm. Scale bar (black) = 200 μ m. (C) Histopathologic examination of vital organs (representative for $n = 4$ biologically independent samples). Scale bar = 200 μ m.

Table R1. Observation of toxic reactions

Clinical observation	Observed sign	Involved system(s)
Respiratory	Dyspnea (abdominal breathing, gasping), apnoea, cyanosis, tachypnea, nostril discharges	CNS, pulmonary, cardiac
Motor activities	Decrease/increase somnolence, loss of righting, catalepsy, ataxia, unusual locomotion, prostration, tremors, fasciculation	CNS, somatomotor, sensory, neuromuscular, autonomic
Convulsion	Clonic, tonic, tonic-clonic, asphyxial, opisthotonos	CNS, neuromuscular, autonomic, respiratory
Reflexes	Corneal, righting, myotact, light, startle reflex	CNS, sensory, autonomic, neuromuscular,
Ocular signs	Lacrimation, miosis, mydriasis, exophthalmos, ptosis, opacity, iritis, conjunctivitis, chromodacryorrhea, relaxation of nictitating membrane	Autonomic, irritation
Cardiovascular signs	Bradycardia, tachycardia, arrhythmia, vasodilation, vasoconstriction,	CNS, autonomic, cardiac, pulmonary
Salivation	Excessive	Autonomic
Piloerection	Rough hair	Autonomic
Analgesia	Decrease reaction	CNS, sensory
Muscle tone	Hypotonia, hypertonia	Autonomic
Gastrointestinal	Soft stool, diarrhoea, emesis, diuresis, rhinorrhea	CNS, autonomic, sensory, GI motility, kidney
Skin	Oedema, erythema	Tissue damage, irritation

Table R2. Body weight at different times and clinical observation results of test animals after treatment

Group	Treatment pathway	Animals number	Sex	Weight(g)				Clinical observation				
				Initial	24h	48h	72h	Immediate	4h	24h	48h	72h
Test group	IV.	231083	Female	18.5	19.4	20.4	21.2	—	—	—	—	—
		231097		19.2	19.8	21.0	22.0	—	—	—	—	—
		231073		19.6	21.3	23.0	24.4	—	—	—	—	—
		231096		18.7	19.5	20.5	21.4	—	—	—	—	—
		231100		18.2	18.6	19.6	20.9	—	—	—	—	—
	IP.	231072	Female	18.7	19.8	20.2	21.1	—	—	—	—	—
		231082		18.3	19.3	20.0	20.7	—	—	—	—	—
		231079		19.9	20.2	20.9	22.0	—	—	—	—	—
		231102		19.0	19.7	20.7	21.6	—	—	—	—	—
		231088		20.8	21.1	22.3	23.2	—	—	—	—	—
Control group	IV.	231084	Female	18.3	19.6	20.4	21.9	—	—	—	—	—
		231074		18.7	19.4	20.4	21.7	—	—	—	—	—
		231103		20.4	21.6	22.6	24.2	—	—	—	—	—
		231086		20.9	21.8	22.7	23.9	—	—	—	—	—
		231104		19.9	21.3	22.1	23.5	—	—	—	—	—
	IP.	231071	Female	19.3	20.1	20.7	22.1	—	—	—	—	—
		231095		20.6	21.5	22.7	23.6	—	—	—	—	—
		231085		18.4	20.0	20.8	21.5	—	—	—	—	—
		231093		19.8	20.4	21.4	22.3	—	—	—	—	—
		231105		21.8	23.1	24.3	25.1	—	—	—	—	—

IV. intravenous injection. IP. intraperitoneal injection. “ - ” : the animal is normal after injection.

Text in the Results section of the manuscript was modified accordingly:

“As demonstrated above, the ex vivo shape memory and recovery properties of PUU-PCL is desirable. Biocompatibility of PUU-PCL was then evaluated before in vivo HIFU experiments. Cell culture experiment showed that PUU-PCL is cytocompatible (Figure S4). Degradation in

PBS is slow as weight loss after 12 w incubation was not significant (Figure S5). Subcutaneous implantation showed that PUU-PCL kept its original shape, did not induce severe inflammation reaction or significantly degrade (Figure S6). Evaluation of the influence of aqueous extract of PUU-PCL on series of physiological functions according to ISO10993 standard showed that PUU-PCL did not induce toxic reactions (Table S1 and S2), further demonstrating its biocompatibility.”

3. Pigs could also be used as well too since their ureters most closely resemble those of humans in size.

Answer: We thank the reviewer for the suggestion. The anatomical structure and relative positions of human, porcine, canine (beagles in our study) and ovine (sheep in our study) bladder, ureter and kidney are similar. The length and diameter of ureters from the model animals vary from those of human. In this case, one only needs to change the length and diameter of the stent to fit the target ureter. On the other hand, we have demonstrated in the canine HIFU experiment that HIFU can focus ultrasound energy at target 5 cm beneath the skin. Clinically, HIFU is used to ablate tumor as deep as 10 cm in patients. In human, the average distance of bladder and kidney to the body surface is shorter than 10 cm. Therefore, theoretically no significant changes to the presented stents or HIFU parameters are required to accommodate for application in patients.

Text in the Results section was modified accordingly:

“...indicating that the HIFU triggered device shape recovery procedure did not impair kidney and liver functions. The anatomical structure and relative positions of human, canine, and porcine bladder, ureter and kidney are similar. The length and diameter of ureters from the model animals vary from those of human. Compared to canine ureter, bladder and kidney, porcine counterparts

generally more closely resemble those of humans in size, thus porcine models could be included in future studies.”

Reviewer 3:

Achieving remotely controlled shape recovery of shape memory instrumentation by HIFU will be revolutionary in applications where medical devices need to be implanted or removed. The authors demonstrate this in simple experiments. The study is overall clearly outlined.

- 1. Please replace citation 18 with the citation below. This is the pioneer study at NEJM that led to the FDA approval of HIFU application in the brain (please note I have no affiliation with the paper NOR am I a co-author of this paper): A Randomized Trial of Focused Ultrasound Thalamotomy for Essential Tremor.**

Answer: We thank the reviewer for introducing the milestone work. We have replaced citation 18 with this work.

- 2. Strictly speaking, image-guided MR thermometry is not precisely real-time. There is an offset of a few minutes depending on the MR sequence. Please replace this section in the paper by indicating that this procedure is nearly real-time or similar.**

Answer: We thank the reviewer and have changed the statement as “The temperature elevation was monitored in nearly real-time using MR thermometry of phase-difference fast spoiled gradient-echo sequences (proton resonant frequency shift method) in the gel pad adjacent to the target sample.”

Text in the Results section was modified accordingly:

“Compared to the two heating options above, high intensity focused ultrasound (HIFU) concentrates ultrasound energy on in vivo targets for efficient heating, while less energy is attenuated in surrounding tissue. In fact, focused ultrasound including HIFU has been widely used, and its safety and efficacy in various applications have been demonstrated²⁰⁻²³. In addition, with

nearly real-time magnetic resonance or ultrasound guidance (MRgHIFU, MRgHIFU; US guided HIFU, USgHIFU), the sound wave energy can be more precisely targeted to minimize off-target damage^{24,25}”

“...The target sample was 7 cm from the ultrasound transducer. Increases in temperature were monitored using nearly real-time MR thermometry of phase-difference fast spoiled gradient-echo sequences (proton resonant frequency shift method) in the gel pad adjacent to the target sample.....”

3. When reporting the MR sequence details, please also write the corresponding SI units.

Answer: The parameters of MR was described in the Methods section. The missing units have been added.

“MR thermometry was performed during each sonication with multiphase multislice echo planar imaging (*FOV/slice thickness/TR/TE/flip angle/echo train length = 28×28 cm/4 mm/210 ms/18.3 ms/35 ° /12*) to monitor approximate changes in temperature during the course of the procedure.”

4. In the finite element model, the authors apply an ultrasound frequency of 1 MHz. The InSightec system operates at a center frequency of 650 kHz. Please explain why 1 MHz was chosen over 650 kHz.

Answer: The HIFU systems used in MRgHIFU and USgHIFU are different. The InSightec system was used for in vitro MRgHIFU experiments, which works at the frequency of 650 kHz (as this equipment is employed more in treatments for bone tumors and skeletal muscle tumors). The Haifu JC system was used in the in vivo USgHIFU experiments, which works at the frequency of 1 MHz.

As evaluating the potential thermal injury in surrounding tissue is one of the important goals of the finite element analysis, we chose 1 MHz, the same frequency used in the animal studies, in simulation.

The 1 MHz ultrasound would have higher efficiency of thermal effect than that of 650 kHz ultrasound in the body (**Figure R7**, from Samanipour R et al: Numerical study of the effect of ultrasound frequency on temperature distribution in layered tissue. *J Therm Biol.* 38 (2013) 287-293). However, 650 kHz ultrasound has deeper penetration through tissue compared to 1 MHz ultrasound. Actually, 650 kHz focused ultrasound could penetrate the skull with less thermal energy deposited in the bone, which was applied in its brain treatment system. In the canine and sheep study, the depths of 1 MHz HIFU penetration were great enough to reach the target devices.

Fig. 12. Temperature rise at focal point over time for different frequencies ($f=0.5$, $f=0.8$, $f=1$, $f=1.5$ and $f=2$ MHz).

Figure R7. The influence of ultrasound frequency on the temperature rise rate. Adapted from Samanipour et al. *J Therm Biol.* 2013.

In addition to the existing simulation results, we simulated HIFU heating of PUU-PCL in a renal pelvis to evaluate heating rate and thermal injury (compared to the model presented in **Figure 6**, we increased the size of PUU-PCL device to reduce direct heating in urine. We exaggerated

thermal injury in **Figure 6** since we attempted to evaluate the safety of HIFU heating more strictly). The results showed that 1 MHz HIFU heated PUU-PCL to above transition temperature in 9 s, whereas 650 kHz HIFU heating took 90 s. Maximum CEM43°C in surrounding tissue was greater in 650 kHz group compared to that in 1 MHz group (**Figure R8**), but both are far below the injury threshold, thus are safe in the simulated condition.

Figure R8. Temperature change induced by HIFU heating and CEM43°C at different locations in the material and in tissue in the bladder after one entire heating-cooling cycle. (A) 650 kHz HIFU, the maximum CEM43°C in tissue is 6.08×10^{-6} min. (B) 1 MHz HIFU, the maximum is CEM43°C in tissue 1.25×10^{-6} min.

Text has been added to the Results section to explain the reason why 1 MHz was chosen:

“Finite element modeling was used to simulate HIFU induced stent heating in the renal pelvis and evaluate the risk of hyperthermia in surrounding kidney tissues. **Since 1 MHz HIFU was used in the canine and sheep studies, 1 MHz HIFU was chosen in the finite element analysis to evaluate risk of potential thermal injury.** The entire simulated polyurethane sample (3 mm diameter solid sphere) was heated above the transition temperature of Fe₃O₄ nanoparticle incorporated PUU-PCL within 25 s (at the end of heating cycle)...”

“A Magnetic resonance guided high intensity focused ultrasound (MRgHIFU) system (ExAblate2000, Insightec, Haifa, Israel) with a phased array was used to induce a PUU-PCL origami. **The Insightec system was equipped with a 650 kHz transducer...**”

“The transducer was driven at the frequency of 1 MHz **as 1 MHz HIFU was used in the animal studies (Type JC USgHIFU, Haifu, China)...**”

“All animal experiments were approved by the Guidelines of Animal Care and Use Committees of Zhejiang University (ZJU20210167). **USgHIFU equipped with a 1 MHz HIFU (JC, Haifu, China) was used in the animal experiments...**”

5. Please describe in detail the HIFU settings that were prescribed including the length (e.g., pulse length, duration of prescription, duty cycle, energy).

Answer: Parameters of HIFU used in the animal studies are listed below, and added to supporting information.

Table R3. HIFU parameters in the animal studies

Model	Mode	Power	Energy	Pulse length	Duration of prescription	Duty cycle
Canine, bladder	line scan	420 W	3360 J	20 mm	8 s	100%
Sheep, liver	line scan	400 W	4000 J/scan	20 mm	10 s/scan	100%

6. For statistics, please indicate in the bar plot figure legend what the star corresponds to (e.g., 0.05%, etc). The statistical analysis section is very scarce. Please elaborate more. If there is no statistician on the team, please include a statistician who can help.

Answer: We thank the reviewer for the comment. We checked the figure legends, and added the missing information in statistical analyses.

Figure legend of **Figure 4** was modified:

*“...Peak load (E) and work (F) required to remove the stents were higher in commercially available stents and PUU-PCL stents with coils compared to their coil-removed counterparts (n = 4 per group). Coil-removed PUU-PCL stent had lower resistance at its smaller diameter. Statistical significance was evaluated using one-way ANOVA with Tukey’s test. Data are presented as means ± SD, *p < 0.05.”*

Legend of Figure R6 is also updated.

*“**Figure R6.** (A) Hepatorenal safety of implanted tubes by blood detection (n = 4). Statistical significance was calculated using one-way ANOVA with Tukey’s test, and data are presented as means ± SEM. ns > 0.05. (B) Histopathologic examination of subcutaneous tissue around the implanted tube with larger magnification insets (representative for n = 4 biologically independent samples). Scale bar (blue) = 2 mm. Scale bar (black) = 200 µm. (C) Histopathologic examination of vital organs (representative for n = 4 biologically independent samples). Scale bar = 200 µm.”*

7. HIFU is not mechanical but thermal in nature. Please remove citation 14.

Answer: We thank the reviewer for the comment, and have modified the text and removed citation 14.

8. In general, citations in the manuscript are very scarce. The authors should revise the citations and include more citations from original research rather than review articles.

Answer: We thank the reviewer for the comment, and have added more citations (the added ones are listed below):

2. Lendlein, A. & Kelch, S. Shape-Memory Polymers. *Angew. Chem. Int. Ed.* **41**, 2034-2057 (2002).

4. Delaey, J., Dubruel, P. & Van Vlierberghe, S. Shape - Memory Polymers for Biomedical Applications. *Adv. Funct. Mater.* **30** 1909047 (2020).

5. Zhao, W., Liu, L., Zhang, F., Leng, J. & Liu, Y. Shape memory polymers and their composites in biomedical applications. *Mater. Sci. Eng. C* **97**, 864-883 (2019).

11. Wang, X., He, Y., Liu, Y. & Leng, J. Advances in shape memory polymers: Remote actuation, multi-stimuli control, 4D printing and prospective applications. *Mat. Sci. Eng. R Rep.* **151**, 100702 (2022).

12. Xia, Y., He, Y., Zhang, F., Liu, Y. & Leng, J. A Review of Shape Memory Polymers and Composites: Mechanisms, Materials, and Applications. *Adv. Mater.* **33**, e2000713 (2021).

13. Li, G., Fei, G., Xia, H., Han, J. & Zhao, Y. Spatial and temporal control of shape memory polymers and simultaneous drug release using high intensity focused ultrasound. *J. Mater. Chem.* **22**, 7692-7696 (2012).

20. Elias, W.J., et al. A Randomized Trial of Focused Ultrasound Thalamotomy for Essential Tremor. *N. Engl. J. Med.* **375**, 730-739 (2016).

21. Razavi, M., et al. Facilitating islet transplantation using a three-step approach with mesenchymal stem cells, encapsulation, and pulsed focused ultrasound. *Stem Cell Res. Ther.* **11**, 405 (2020).
22. Ren, T., Steiger, W., Chen, P., Ovsianikov, A. & Demirci, U. Enhancing cell packing in buckyballs by acoustofluidic activation. *Biofabrication* **12**, 025033 (2020).
23. Xu, Y., Yang, L. & Wong, F. *Surgical techniques of focused ultrasound ablation in benign uterine diseases*. (Springer Singapore, 2023).
33. Tanaka, M., et al. A Closed-loop Transcutaneous Power Transmission System with Thermal Control for Artificial Urethral Valve Driven by SMA Actuator. *J. Intell. Mater. Syst. Struct.* **17**, 779-786 (2006).
34. Liao, Y., et al. Magnetically controlled artificial urinary sphincter: An overview from existing devices to future developments. *Artif. Organs* **47**, 1075-1093 (2023).
35. Marziale, L., et al. Artificial Sphincters to Manage Urinary Incontinence: A Review. *Artif. Organs* **42**, E215-E233 (2018).
36. Hached, S., et al. Novel, Wirelessly Controlled, and Adaptive Artificial Urinary Sphincter. *IEEE/ASME T. Mech.* **20**, 3040-3052 (2015).

REVIEWERS' COMMENTS

Reviewer #1 (Remarks to the Author):

The authors replied all the proposed comments adequately and revised the manuscript. I have no more concerns about it. It could be accepted for publication.

Reviewer #2 (Remarks to the Author):

The revisions carried out are outside my area of expertise. The main foundation of what this technology would allow us to do medically is unchanged in the manuscript and i think still very interesting and show great promise.

Reviewer #3 (Remarks to the Author):

I thank the authors for their diligent work in revising the manuscript.

My only comment is to please remove "nearly" from this sentence:

"nearly real-time magnetic resonance or ultrasound guidance ..."

I was only referring to MR thermometry. The authors have indicated that MR thermometry is nearly real-time. Alternatively, the authors can continue using real-time MR thermometry but explain in the methods that there was a delay of X minutes for acquiring thermometry data.

I recommend the manuscript for publication.

Thank you.